# In vivo spontaneous Ca²⁺ activity in the pre-hearing mammalian cochlea

Francesca De Faveri [1], Federico Ceriani [1] ✉ & Walter Marcotti [1,2] ✉

The refinement of neural circuits towards mature function is driven during development by patterned spontaneous calcium-dependent electrical activity. In the auditory system, this sensory-independent activity arises in the pre-hearing cochlea and regulates the survival and refinement of the auditory pathway. However, the origin and interplay of calcium signals during cochlear development is unknown in vivo. Here we show how calcium dynamics in the cochlear neuroepithelium of live pre-hearing mice shape the activity of the inner hair cells (IHCs) and their afferent synapses. Both IHCs and supporting cells (SCs) generate spontaneous calcium-dependent activity. Calcium waves from SCs synchronise the activity of nearby IHCs, which then spreads longitudinally recruiting several additional IHCs via a calcium wave-independent mechanism. This synchronised IHC activity in vivo increases the probability of afferent terminal recruitment. Moreover, the modiolar-to-pillar segregation in sound sensitivity of mature auditory nerve fibres appears to be primed at pre-hearing ages.

In mammals, the processing of sensory information, including those in the auditory and visual systems, relies on the precise formation of the underlying neural circuits. The origin and development of sensory circuits require genetic programs, with the initial wiring being determined by axon guidance molecules[1–3]. However, the refinement of these circuits towards functional maturity depends on spontaneous electrical activity that originates within the immature sensory organs independently from any external stimulus[4,5]. In the developing auditory system, periodic bursts of spontaneous electrical activity are present in neurons of the ascending auditory pathway[6–9], which have been shown to be required for the survival of nascent neurons and for the refinement of the neural circuitry[10–13]. Evidence supporting the cochlear origin of this spontaneous firing activity comes from experiments showing that its ablation or block of the peripheral firing activity during development abolishes the firing in the auditory centres[6,14], and results in substantial loss of cochlear nucleus neurons[15,16]. Cochlear activity propagates to the auditory afferent fibres via Ca²⁺ mediated release of glutamate from the primary sensory receptor inner hair cells (IHCs)[8,17].

Ex vivo experiments have shown that IHCs are capable of firing spontaneous action potentials before the onset of hearing[18–21], which occurs at around postnatal day 12-13 in most altricial rodents[22,23]. IHC action potentials are primarily driven by the interplay between an inward Ca²⁺ current and a slowly activating delayed rectifier outward K⁺ current[18–20,24], and are modulated by the neurotransmitter ACh released from the efferent fibres that transiently innervate pre-hearing IHCs[25,26]. However, other studies have proposed that IHCs are intrinsically silent and instead it is the spontaneous release of ATP from glial-like supporting cells that, by activating a P2Y-mediated signalling cascade, leads to the accumulation of K⁺ in the intercellular space, IHC depolarisation and subsequent firing activity in the afferent fibres[27,28]. Despite this large body of work, our understanding of the mechanisms responsible for the initiation and modulation of the firing activity in the cochlea and its dynamics in vivo is still limited and much debated. This is primarily dictated by the fact that experiments have been performed using cochlear explants or organotypic cultures that, although valuable, completely disrupt the normal innervation and physiology of the mammalian cochlea. This includes the separation of the different solutions (perilymph and endolymph) surrounding the sensory epithelium, which impact on the function of both hair cells and supporting cells, the presence of an endocochlear potential and the efferent feedback from the brainstem[29–31].

[1]School of Biosciences, University of Sheffield, Sheffield S10 2TN, UK. [2]Neuroscience Institute, University of Sheffield, Sheffield S10 2TN, UK. ✉e-mail: f.ceriani@sheffield.ac.uk; w.marcotti@sheffield.ac.uk

In this study, we have established a surgical approach that allows us to optically access the cochlear sensory epithelium in live mice. This has enabled us to investigate spontaneous activity in the intact cochlea with single-cell resolution using mice expressing the genetically encoded Ca$^{2+}$ indicator GCaMP6 in the IHCs, afferent fibres or surrounding supporting cells. We found that, in vivo, IHCs display spontaneous Ca$^{2+}$ transients both independently and synchronised to nearby Ca$^{2+}$ waves. Although the synchronised activity of IHCs was initiated by the spontaneous Ca$^{2+}$ waves, its longitudinal spread affecting up to several IHCs occurred via a faster Ca$^{2+}$ wave-independent mechanism. Moreover, we showed that the majority of spiral ganglion neuron (SGN) afferent terminals innervating individual IHCs only became simultaneously activated during the synchronised Ca$^{2+}$ activity of several IHCs. The coordination of afferent activity is an essential mechanism for the proposed functional refinement of tonotopic maps in the auditory pathway. Finally, we have shown that the afferent terminals appear to establish the functional segregation according to their position around individual IHCs, which is a key characteristic of adult neurons, already at pre-hearing stages of development.

## Results

### Surgical approach to access the cochlear sensory epithelium in vivo

We have established a surgical approach to optically access the sensory epithelium in the intact mammalian cochlea of live mice. Anaesthetised mice were positioned on a heating mat (belly facing upward) to maintain a stable body temperature throughout the non-recovery in vivo procedure. After cutting the skin with fine scissors, the muscles and connective tissue on the side of the trachea were gently separated to reach the bulla (Fig. 1a), which houses the ossicles of the middle ear. Although the bulla encloses an air-filled cavity in post-hearing mice (Fig. 1b), during pre-hearing stages of development it is filled with a gelatinous material that was removed with forceps to be able to visualise the cochlear apical coil (Fig. 1c). After exposing the cochlea (Fig. 1c), a small part of the bone covering the region that corresponds to the 8–18 kHz region in hearing mice[32] was gently removed with fine forceps (Fig. 1d). The space above the cochlea was then filled with an extracellular solution at 37 °C (see Methods), which allowed us to visualise the cochlear sensory epithelium with water-immersion objectives (Fig. 1e-g). Crucially, the surgical procedure was performed without breaking the lateral wall membranes sealing the cochlear partition. This prevented any disruption, or mixing, of the endolymph and perilymph solutions in the different cochlear compartments, and preserved the endocochlear potential (EP), which is known to develop from about the end of the first postnatal week in mice[33] and is crucial for mechanoelectrical transduction and normal hair cell physiology[34]. Indeed, we found that the EP measured in pre-hearing mice that underwent the above surgical procedure (Fig. 1a–g) was preserved and significantly increased with age (P7: 15.8 ± 0.8 mV, $n = 6$; P10: 27.2 ± 2.7 mV, $n = 3$, $P < 0.0001$, $t$-test). The size of the EP was comparable to that previously measured in age-matched mice anaesthetised using ketamine and xylazine[33], which are widely used to assess normal hearing function. This experimental approach, combined with the use of mice expressing genetically encoded Ca$^{2+}$ indicators in the primary sensory receptors, the inner hair cells (IHCs), their afferent terminals or surrounding supporting cells, was used to investigate spontaneous activity in the developing cochlea with single-cell and single-synapse resolution using two-photon imaging.

### Spontaneous calcium signals in developing IHCs from live mice

Calcium signals in developing IHCs were investigated by crossing GCaMP6f floxed mice (GCaMP6f$^{fl/fl}$) with Myo15-Cre$^{+/-}$ or Atoh1-Cre$^{+/-}$ mice (see Methods). Under in vivo experimental conditions (Fig. 1a–g), IHCs exhibited rapid spontaneous Ca$^{2+}$ transients throughout pre-

hearing stages of development (Fig. 1h–j, Fig. 2a; Supplementary Fig. 1a; Supplementary Movie 1). These Ca$^{2+}$ transients were present for the full duration of the recordings, which for ethical reasons was set to a maximum of about 1 h (Supplementary Fig. 2). The average frequency of Ca$^{2+}$ transients in pre-hearing IHCs (P3-P10: 0.99 ± 0.83 events/min, Fig. 2a) was age dependent ($P < 0.0001$, Kruskal-Wallis), it decreased during the second postnatal week and was no longer detected from P12 onwards. The Ca$^{2+}$ transient full duration at half maximum was on average 1.50 ± 0.75 s (Fig. 2b). Although Ca$^{2+}$ transients were present in IHCs as single events separated by long periods of quiescence, IHCs also showed periods of burst-like activity lasting several seconds (Fig. 2c). To demonstrate that Ca$^{2+}$ transients observed in vivo depend on Ca$^{2+}$ entry through L-type (Ca$_V$1.3) Ca$^{2+}$ channels, which carry about 90% of the Ca$^{2+}$ current in pre-hearing cells[24,35], we performed some in vivo recordings using Cav1.3 knockout mice (Cav1.3$^{-/-}$). Since this mouse line does not constitutively express GCaMP, we injected AAV-PHP.eB-jGCaMP8m directly into the cochlea of newborn mice in vivo. We found that Ca$^{2+}$ transients were completely abolished in the large majority of IHCs recorded in vivo from P8-P10 Cav1.3$^{-/-}$ mice compared to control mice ($P < 0.0001$, Mann-Whitney U-test, Fig. 2d; Supplementary Fig. 1b).

To provide a direct comparison between IHC Ca$^{2+}$ transients recorded in vivo and ex vivo[27,36,37], we performed recordings from cochlear explants of P7-P8 GCaMP6f$^{fl/fl}$Myo15-Cre$^{+/-}$ mice with the same imaging setup used for the above in vivo work (Fig. 2a–c). We found that Ca$^{2+}$ transients recorded from 223 IHCs under ex vivo conditions had a lower frequency compared to those recorded in vivo ($P < 0.0001$, Mann-Whitney U-test, Fig. 2e, f; Supplementary Fig. 1c). We also found that in about 41% of the IHCs from cochlear explants (104 out of 258 IHCs, 12 mice) Ca$^{2+}$ transients became progressively more frequent, leading to a sustained Ca$^{2+}$ signal that started from an average of 6.1 min and lasted until the end of the recording (up to 15 min) (Fig. 2e). This sustained high-Ca$^{2+}$ level, which was never observed in vivo, indicates that under ex vivo conditions IHCs are likely to become Ca$^{2+}$ overloaded and unhealthy.

Ex vivo experiments have shown that Ca$^{2+}$ transients in developing hair cells underlie bursts of Ca$^{2+}$-dependent action potentials[19,27,38,39]. Extrapolation of spiking activity from in vivo Ca$^{2+}$ imaging recordings (Fig. 2g) using the deep-learning-based tool CASCADE[40] yielded an average firing rate of 0.27 Hz (776 IHCs, 28 mice; Fig. 2h). This frequency was comparable to that previously measured using cell-attached patch clamp from ex vivo cochlear IHCs during the first postnatal week (0.26 Hz)[19], a time when the EP is not yet established. The average maximal firing rate of the extrapolated Ca$^{2+}$ transients was 14 Hz (Fig. 2i).

On average 58% of Ca$^{2+}$ transients observed in an IHC occurred nearly simultaneously in more than 3 neighbouring IHCs (Fig. 3a–c; Supplementary Fig. 3). This coordinated activity spread along the longitudinal axis of the cochlea (Fig. 3a, b; Supplementary Movie 2), and it is most likely initiated by the propagation of ATP-induced Ca$^{2+}$ waves, which have been shown to originate in the surrounding supporting cells in cochlear explants or organotypic cultures[14,27,28,41]. On average, about 7 IHCs were recruited during coordinated Ca$^{2+}$ transients (up to about 20 IHCs were sometimes involved, Fig. 3d; Supplementary Fig. 1d) resulting in a high degree of correlation between the activity of nearby IHCs (Fig. 3e), which decreased exponentially with the distance between cells (Fig. 3f). In about 62% of the events involving coordinated Ca$^{2+}$ activity of more than 3 IHCs, between 1 and 4 cells within the activated cohort had substantially reduced, or failed to elicit any Ca$^{2+}$ transients ("skipped"). These skipped IHCs were not silent since they were able to elicit Ca$^{2+}$ transients before and/or following the coordinated activity (Supplementary Fig. 4). Although the reason for these observations is currently unknown, it is possible that the skipped IHCs were transiently hyperpolarised by the inhibitory efferent system that innervates pre-hearing IHCs[25]. Finally, we found

that about 27% of the total Ca²⁺ transients in IHCs occurred as spontaneous uncoordinated events independent from neighbouring cells (Fig. 3c), the existence of which has been debated based on recordings from cochlear explants[14,19,20,27,28,37,42].

## In vivo spontaneous calcium waves in developing supporting cells

Ex vivo studies have shown that spontaneous Ca²⁺ waves in supporting cells of the greater epithelial ridge (GER: Fig. 4a) are required for the coordination of activity in multiple IHCs[27,28]. These Ca²⁺ waves, analogous to those found in glial cells[43], are due to the spontaneous

release of ATP from supporting cells[27,44], which activates G-protein-coupled P2Y autoreceptors located on their apical surface[45–48]. Activation of these receptors initiates a signalling cascade that causes mobilisation of Ca²⁺ from intracellular stores[49], further release of ATP and K⁺ release, which then depolarises the IHCs[28]. Calcium waves can also be triggered by damage to the sensory epithelium[50,51], limiting the suitability of acute cochlear explants for studying these phenomena.

By imaging the cochlea of mice expressing GCaMP6 in supporting cells (*GCaMP6f^{fl/fl}Pax2-Cre^{+/−}*), we confirmed that Ca²⁺ waves do occur spontaneously in the supporting cells of the GER in vivo

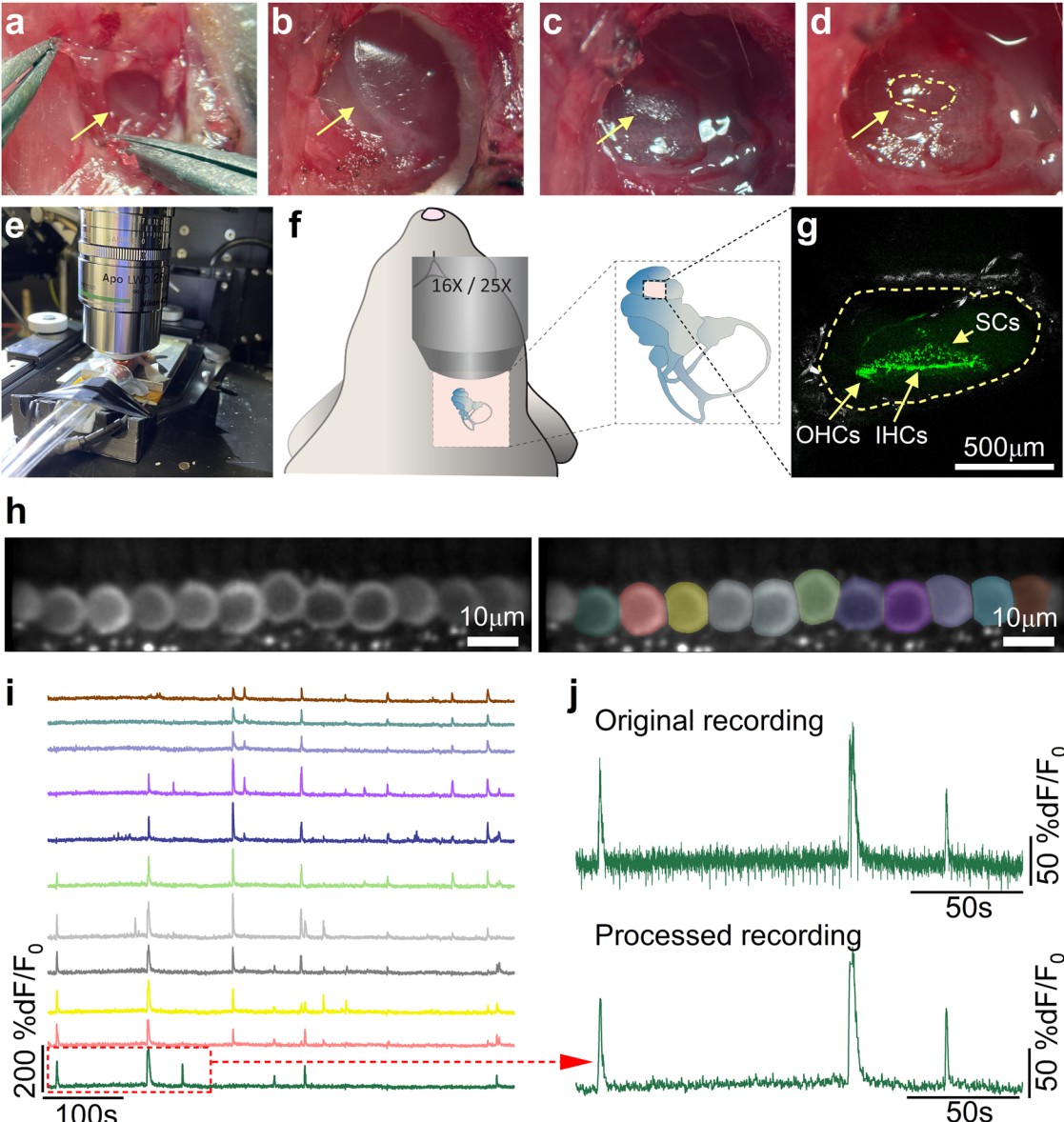

**Fig. 1 | In vivo imaging of spontaneous Ca²⁺ signals in cochlear inner hair cells.** Surgical (**a–d**) and imaging (**e–g**) experimental conditions used for in vivo cochlear imaging in live mice. The surgical procedure allowed localisation (**a**) and exposure (**b**) of the bulla, which was removed to view the apical coil of the cochlea (**c**). Here, a small opening of the bony structure (**d**, dashed line) was performed, but without breaking the underneath membranes sealing the cochlear partition. After filling the space above the cochlea with extracellular solution at 37 °C, the mice were moved to the microscope stage for imaging (**e, f**). Recordings were made from the cochlear sensory epithelium where inner hair cells (IHCs) and supporting cells (SCs) could be visualised using conditional expression of GCaMP6f (**g**: P4, *GCaMP6f^{fl/fl}Atoh1-Cre^{+/-}* mouse). **h** Left: average

intensity projection of a timelapse recording highlighting GCaMP6f expression in the IHCs from the apical coil of a P6 *GCaMP6f^{fl/fl}Myo15-Cre^{+/-}* mouse. Right: ROIs generated using a semi-automated identification approach (see Methods), which were used to measure spontaneous Ca²⁺ signals from individual IHCs from the left panel. **i** Fluorescence time series computed as pixel-averages from the ROIs in **h**, highlighting spontaneous Ca²⁺ activity in IHCs colour-matched to panel (**h**). **j** Comparison of a representative fluorescence trace from an IHC (red box in panel **i**) before (top) and after (bottom) the post-processing of the acquired 2-photon image timelapse. Note the motion artifacts in the original recording (downward trace deflections) due to the breathing of the mouse. See Methods for the recording adjustment procedure.

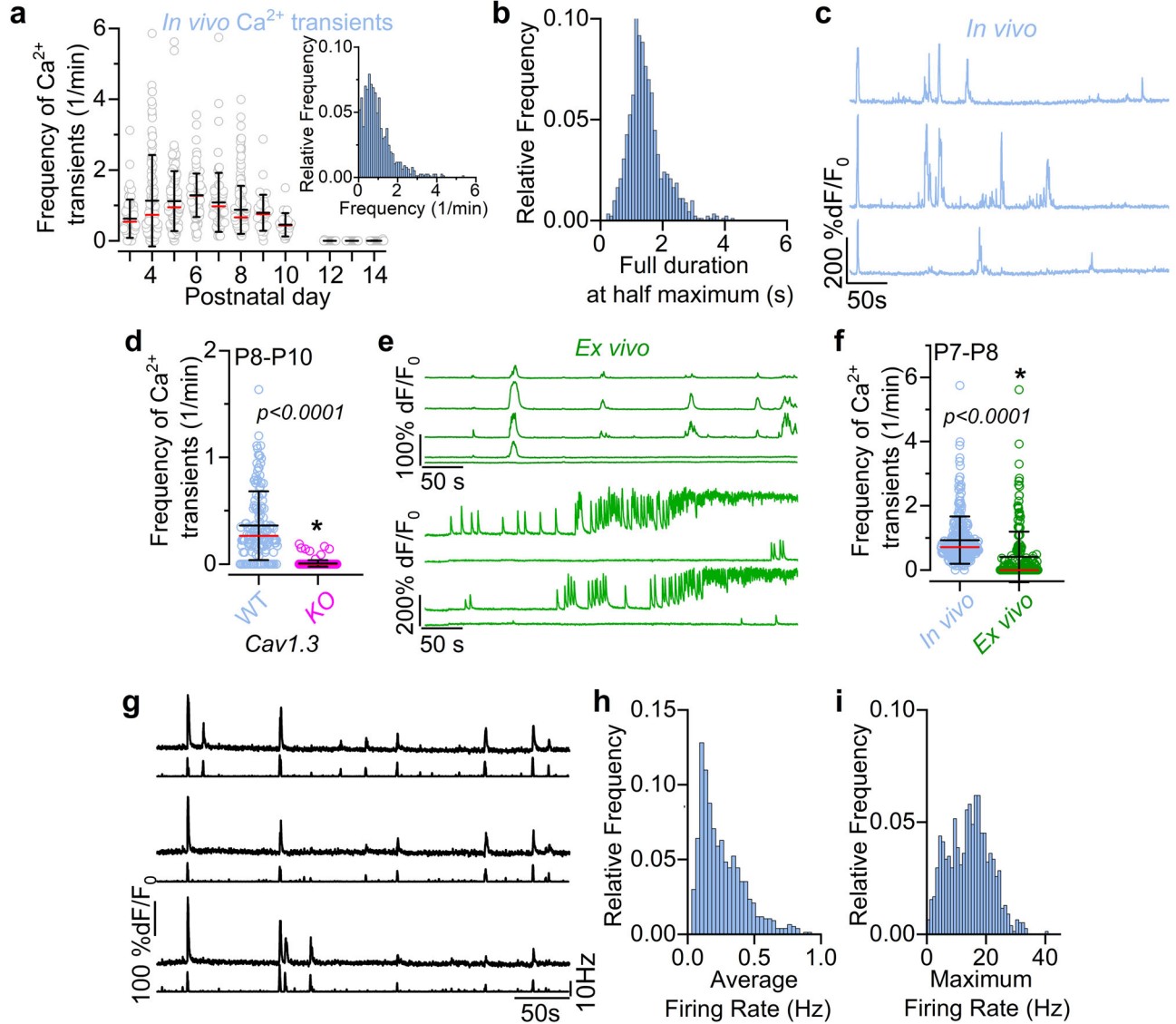

**Fig. 2 | Biophysical characteristics of spontaneous Ca²⁺ activity in developing IHCs. a** Frequency of spontaneous Ca²⁺ transients in IHCs as a function of postnatal development day. Number of IHCs (open circles) from left to right: 73,128,138,96,83,195,41,22,71,47,82 from 34 mice (P3-P6: *GCaMP6f^fl/fl Atoh1-Cre^+/-* mice; P4-P14: *GCaMP6f^fl/fl Myo15-Cre^+/-* mice). This data set includes all IHCs that had at least 5 minutes of recording. Inset: histogram showing the distribution of frequency of spontaneous Ca²⁺ transients present in developing IHCs. **b** Histogram showing the distribution of the full duration at half-maximum of Ca²⁺ transients from all 917 IHCs between P3 and P10. **c**, Examples of Ca²⁺ signals in IHCs showing both single and burst-like Ca²⁺ transients. **d** Frequency of spontaneous Ca²⁺ transients in P8-P10 IHCs from wild-type (WT, blue) and *Ca_V1.3* knockout (*KO: Ca_V1.3^-/-*, magenta) mice. Since these mice do not constitutively express GCaMP, newborn pups were transduced in vivo with AAV-PHP.eB-jGCaMP8m. Ca²⁺ signals were identified in 117 wild-type IHCs (out of 142 IHCs, 6 mice), but only in 9 *Ca_V1.3^-/-* IHCs

(out of 179 IHCs, 7 mice). **e** Fluorescence time series from GCaMP-expressing IHCs from two cochlear explants (ex vivo), highlighting spontaneous Ca²⁺ activity. **f** Comparison of the frequency of Ca²⁺ transients recorded from IHCs of P7-P8 mice under in vivo conditions (blue, 278 IHCs, 8 mice) and from cochlear explants (ex vivo, green: 223 IHCs, 12 mice). In the IHCs where Ca²⁺ signals developed into a sustained elevation, the frequency of Ca²⁺ transients was estimated up to the onset of the sustained activity. **g** Expanded view of selected fluorescence traces from Fig. 1i shown above the inferred Ca²⁺ action potential firing rate (bottom traces) extrapolated using the CASCADE deep learning toolbox (see Methods). Histograms showing the distributions of estimated average (**h**) and maximum (**i**) firing rate of Ca²⁺ spikes obtained using the CASCADE toolbox during development. Data are from 776 IHCs and 28 mice (P3-P10). Average data are shown as mean ± SD (median: red lines). Statistical tests in panels **d**, **f** are from two-sided Mann-Whitney *U*-test.

(Fig. 4b, c; Supplementary Movie 3). The average duration of individual Ca²⁺ waves was 2.24 ± 1.00 s (Fig. 4d), which was slower than that of Ca²⁺ transients in IHCs (Fig. 2b, *P* < 0.0001, Mann-Whitney *U*-test). Our in vivo findings also show that the frequency of Ca²⁺ waves did not significantly change throughout pre-hearing stages of development (*P* = 0.0580, one-way ANOVA, Fig. 4e; Supplementary Fig. 1e), while their area progressively decreased over time (*P* < 0.0001, Kruskal-Wallis, Supplementary Fig. 5). This contrasts with ex vivo data showing a large increase in the frequency and maximal area of Ca²⁺ waves with postnatal development[8]. Both the mean expansion

and contraction speed of the Ca²⁺ waves also decreased over time (*P* < 0.0001, for both comparisons, Kruskal-Wallis, Fig. 4f, g; Supplementary Fig. 1f, g). As seen for Ca²⁺ transients in IHCs (Fig. 2a), Ca²⁺ waves in live mice were almost completely abolished by P12 (Fig. 4e). Although in the P14 cochlea we observed Ca²⁺ transients occurring in individual supporting cells in the proximity of IHCs, these remained confined and failed to develop into intercellular Ca²⁺ waves (Supplementary Movie 4).

To provide a better understanding of the different Ca²⁺ dynamics between the Ca²⁺ waves recorded in vivo and those using alternative

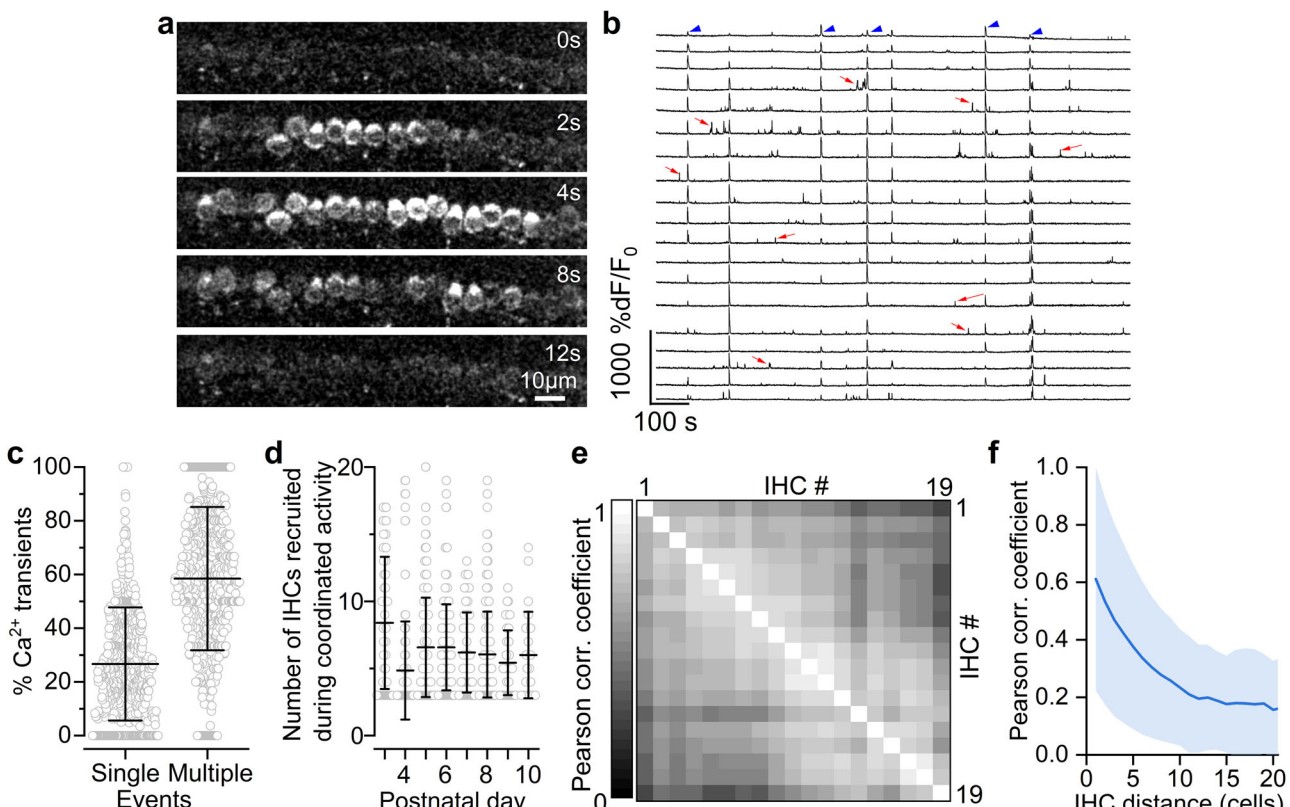

**Fig. 3 | Inner hair cells display both independent and coordinated spontaneous calcium activity in vivo. a** Still images from a timelapse recording from a P8 *GCaMP6f^{fl/fl}Myo15-Cre^{+/-}*mouse, displaying the propagation of spontaneous Ca²⁺ activity across multiple nearby IHCs. Each image represents an average of 30 frames from the original recording. **b** Fluorescence traces highlighting Ca²⁺ activity in a 20-minute recording from 19 IHCs from a P4 mouse. Traces represent Ca²⁺ activity from adjacent IHCs, some of which occurred simultaneously in several cells (arrowheads). These coordinated events appeared alongside less prominent uncorrelated, single-cell transients (arrows). **c**, Percentage of events with Ca²⁺ transients in "single" or "multiple" ( > 2 IHCs) events. Note that 2-cell events (15%) were excluded to minimise any misrepresentation of random coincidental events. Each datapoint represents an individual IHC (P3-P10, 740 active IHCs out of 776, 28 mice). Average data are shown as mean ± SD. **d** Number of IHCs involved in coordinated activity as a function of age. Each data point represents a multicellular event (i.e. Ca²⁺ transients occurring simultaneously in >2 IHCs). Number of events from left to right: 86, 157, 295, 200, 105, 347, 47, 23 from the P3-P10 IHCs in Fig. 2a (31 mice). Data are shown as mean ± SD. At all ages tested, coordinated Ca²⁺ events were observed in up to about 20 cells simultaneously; this value could be slightly underestimated because in a few recordings the longitudinal spread of the Ca²⁺ transients exited the field of view. **e** Correlation matrix computed from the Ca²⁺ traces in panel **b**. Coordinated activity resulted in a high degree of correlation between nearby IHCs. **f** Average correlation coefficient (solid line: average Pearson correlation coefficient; shaded area: standard deviation) as a function of the distance between IHCs.

approaches, we used the same *GCaMP6f^{fl/fl}Pax2-Cre^{+/-}* mice and performed experiments either in vivo, but following the rupture of the membrane sealing the cochlear partitions and thus creating an unphysiological environment (Fig. 4h, i), or from cochlear explants (Fig. 4j, k). Under both unphysiological conditions, we found that the maximum area and duration at half maximum of the spontaneous Ca²⁺ waves were significantly increased compared to those measured in vivo ($P < 0.0001$ for all comparisons with the in vivo values, Dunn post-test, Kruskal-Wallis, Fig. 4l,m; Supplementary Fig. 1h, i). The different Ca²⁺ wave dynamics under unphysiological recording conditions were also highlighted by the multimodal distribution of their durations, highlighting the presence of longer Ca²⁺ waves not observed in vivo (Supplementary Fig. 6). It is possible that the largest Ca²⁺ waves recorded ex vivo or when the cochlear integrity was compromised are caused by cochlear damage signals rather than physiological responses. Interestingly, despite the use of anaesthetic, the changes in Ca²⁺ dynamics recorded in vivo following the rupture of the membrane sealing the cochlear partitions were remarkably similar to those from cochlear explants. This observation, together with the comparable EP between our recordings (using the anaesthetic isoflurane) and those obtained with the commonly used anaesthetic ketamine-xylazine, indicates that isoflurane is likely to have no or negligible effect on Ca²⁺ signals in the developing cochlea.

## Functional interaction between supporting cells and IHCs in vivo

Since *Pax2-Cre* drives recombination in most of the cells of the inner ear, including the IHCs (see Methods), we investigated the spatial relationship between spontaneous activity in the GER and IHCs in a subset of recordings where the latter were clearly distinguishable from the fluorescent background of the enclosing supporting cells (Fig. 5a, Supplementary Movie 5). We found that the large majority of Ca²⁺ waves occurred within 15 μm from the sensory cells (Fig. 5b), and primarily expanded in the longitudinal direction (i.e., along the coil axis of the cochlea, Fig. 5c). About 31% of the total number of Ca²⁺ waves originating within 35 μm from the IHCs were able to reach the sensory cells and elicit synchronised Ca²⁺ signals. When only considering the subset of Ca²⁺ waves that reached the IHC area, about 52% were able to coordinate the activity of the sensory cells. Although the reason for having a large number of Ca²⁺ waves failing to generate Ca²⁺ signals in the IHCs is currently unclear, it is possible that feedback from the inhibitory cholinergic efferent system, which is known to innervate the pre-hearing IHCs[25,26] and is preserved in our in vivo conditions, keeps IHCs hyperpolarised and thus unable to respond to the Ca²⁺ waves.

Although the IHCs in close proximity of a Ca²⁺ wave were the first to respond with Ca²⁺ transients, the total number of IHCs activated

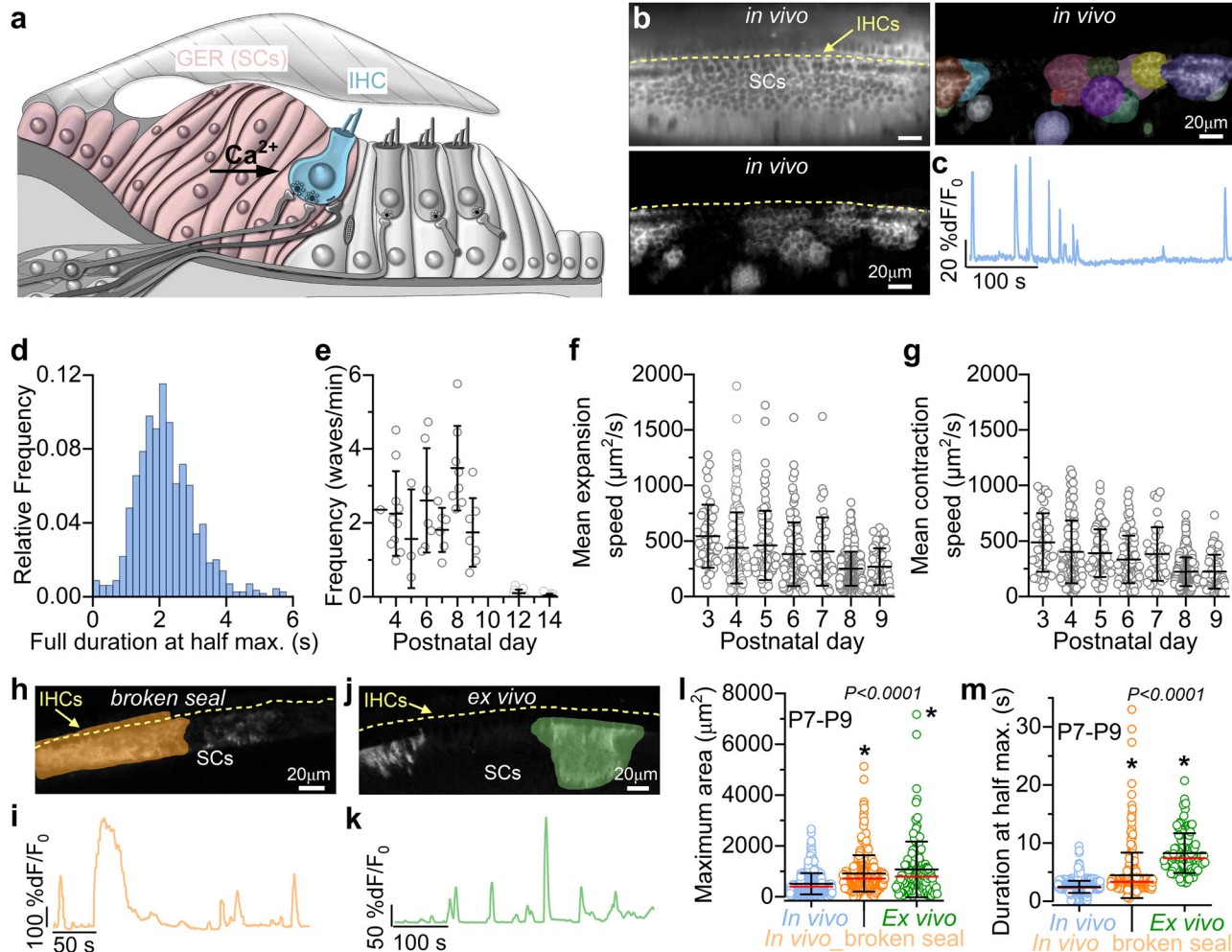

**Fig. 4 | Calcium waves in supporting cells in live mice. a** Diagram showing a cross-section of an immature Organ of Corti. **b** Top left: average intensity projection displaying GCaMP6f expression in the cochlear epithelium from a P4 *GCaMP6f^fl/fl^Pax2-Cre^+/-^* mouse. Bottom left: maximal intensity projection, highlighting $Ca^{2+}$ activity in the supporting cells during a 10-minute recording. Right: semi-automatic ROI segmentation. **c** Fluorescence trace from a square ROI drawn across the entire GER. **d** Histogram of the full duration at half maximum of $Ca^{2+}$ waves (1145 $Ca^{2+}$ waves, 20 mice). **e** Average frequency of $Ca^{2+}$ waves during pre-hearing stages of development ($Ca^{2+}$ waves/min). Each data point represents one recording from 33 mice (number of recordings from left to right: 1,10,3,7,6,8,7,18,21). Mean $Ca^{2+}$ wave expansion (**f**) and contraction (**g**) speed as a function of age. Number of $Ca^{2+}$ waves from left to right for each panel: 47, 151, 86, 108, 41, 399, 77 from 20 mice. Top panels **h** and **j**: average intensity projection displaying GCaMP6f signals in the cochlear epithelium from *GCaMP6f^fl/fl^Pax2-Cre^+/-^* mice at P8 (**h**, broken seal, orange: in vivo but with the exposed cochlear duct following the rupture of the membrane sealing the cochlear partition Fig. 1d) and P7 (**j**, ex vivo, green: cochlear explant). Bottom panels **i** and **k**: corresponding fluorescence traces from the ROI drawn across the GER in panels **h** and **j**, respectively. **l, m** Comparison of the maximum area (**l**) and full duration at half maximum (**m**) of $Ca^{2+}$ waves recorded from P7-P9 cochleae under in vivo (blue: 621 $Ca^{2+}$ waves (**l**), 614 $Ca^{2+}$ waves (**m**), 10 mice), in vivo but with the broken membrane (orange, 329 $Ca^{2+}$ waves (**l**), 324 $Ca^{2+}$ waves (**m**), 6 mice) and ex vivo (green: 135 $Ca^{2+}$ waves (**l**), 120 $Ca^{2+}$ waves (**m**), 3 mice) conditions. Average data are shown as mean ± SD (median: red lines). Note that the different number of $Ca^{2+}$ waves between panel (**l**) and (**m**) is because some $Ca^{2+}$ waves start or end outside the recording time and their duration could not be mesured. Statistical tests in panels **l, m** are from Kruskal-Wallis with Dunn's post-test.

often exceeded the longitudinal extension of the corresponding $Ca^{2+}$ wave in the GER (Fig. 5d, e), similarly to previous reports[47]. This meant that a $Ca^{2+}$ wave of average longitudinal extension (34 ± 13 μm, 116 waves) led to the activation of about 8 IHCs, spanning a significantly longer length of the cochlear epithelium (87 ± 56 μm, $P < 0.0001$, Mann Whitney U-*test*, Fig. 5e). The long-distance propagation of spontaneous $Ca^{2+}$ transients among the IHCs was not accompanied by the spread of $Ca^{2+}$ waves, since there was no overt $Ca^{2+}$ increase in the supporting cells in between the active IHCs (inner phalangeal cells, IPCs) outside the area covered by the wave originating in the GER (Fig. 5a,d, Supplementary Fig. 7). Moreover, the lateral propagation of spontaneous activity amongst the IHCs was about 3 times faster (61 ± 38 μm/s) than the longitudinal spread of the $Ca^{2+}$ waves in the GER (22 ± 8 μm/s, $P < 0.0001$, Mann Whitney U-*test*, Fig. 5f). This suggests that an alternative mechanism to that previously proposed in cochlear explants[8,27]

is likely to be responsible for the depolarisation and propagation of $Ca^{2+}$ signals across multiple IHCs.

## Calcium signals in the afferent terminals from the cochlea of live mice

IHCs are innervated by type I spiral ganglion neurons (SGNs), the terminals of which have been shown to segregate around the IHCs basolateral membrane in adult rodents[52–55], a feature that may reflect the different thresholds and spontaneous rate characteristics recorded in these fibres[52,56,57].

Calcium signal dynamics in the maturing SGN terminals around the IHCs were investigated using either *Snap25-GCaMP6s* or *GCaMP6f^fl/fl^NeuroD1-Cre^+/-^* mice. Two-photon imaging allowed us to visualise synaptic terminals at the targeted focal plane (see Methods), capturing the activity of 1 to 10 semi-automatically segmented bright hot-spots

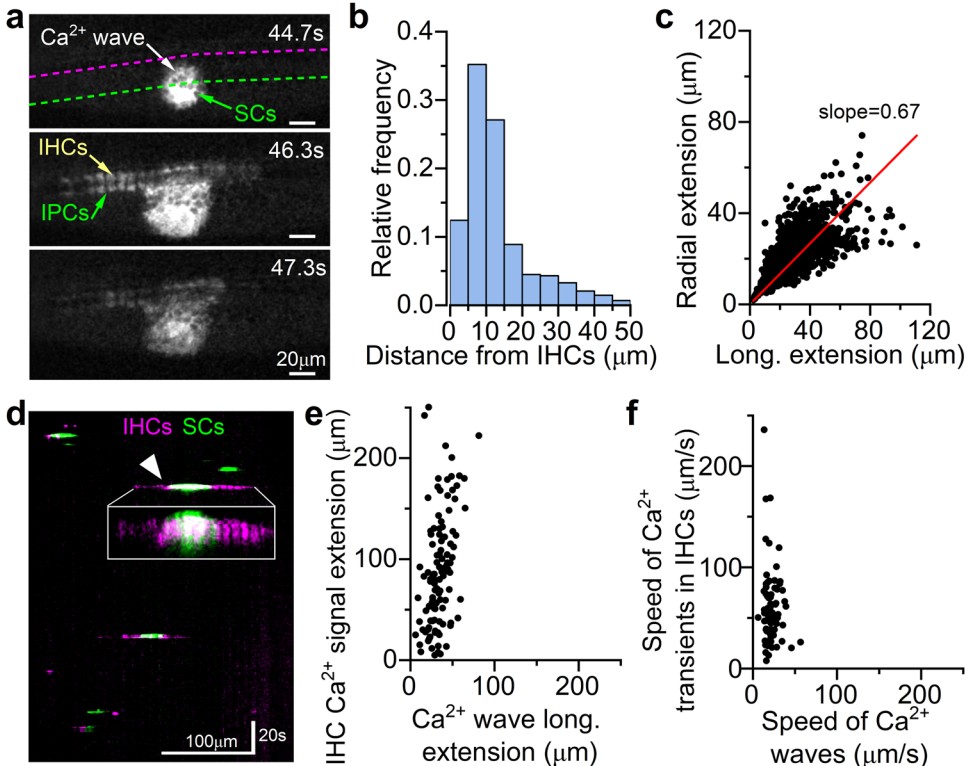

**Fig. 5 | Simultaneous calcium activity in supporting cells and IHCs in live mice. a** Images from a P4 mouse showing the propagation of spontaneous Ca²⁺ waves from the GER to inner hair cells (IHCs). IPCs: inner phalangeal cells located in between the IHCs. SCs: supporting cells within the GER. **b** Histogram showing the number of Ca²⁺ waves as a function of their distance (position of the centroid at their maximal expansion) relative to the IHCs (1159 Ca²⁺ waves, 20 mice). **c** Scatterplot showing the relationship between longitudinal and radial extension of the Ca²⁺ waves from panel **b** measured at their maximal extension. **d** Kymograph obtained by drawing lines across the IHCs and GER regions (magenta and green lines, respectively, in panel **a**). The inset shows a magnification of the Ca²⁺ wave

indicated by the arrowhead, highlighting the larger longitudinal spread of the Ca²⁺ signal in the IHCs compared to that of the GER. Note the absence of Ca²⁺ signals in the IPCs. **e** Scatterplot showing the relationship between the spread of Ca²⁺ waves in the supporting cells of the GER and Ca²⁺ signals in the IHCs from a subset of recordings from panel **b** where both events could be measured simultaneously (116 events). **f** Scatterplot showing the relationship between the speed of longitudinal propagation of Ca²⁺ activity across IHCs and that of the Ca²⁺ waves in the GER (75 events: the speed of Ca²⁺ waves could not be measured when initiated simultaneously to the mouse breathing).

per IHC identified as SGN terminals (average: 4 ± 2 SGN terminals per IHC, see Methods for image segmentation and assignment criterion to individual IHCs). Spontaneous Ca²⁺ transients in SGN afferent terminals were evident on both the modiolar (towards the cochlear nerve) and pillar (towards the outer hair cells) sides of pre-hearing IHCs (Fig. 6a, b, Supplementary Movie 6). Similar to presynaptic IHC activity, we observed Ca²⁺ transients in SGN terminals spanning several IHCs and causing the coordinated activation of large numbers of afferent terminals (Fig. 6b, c, Supplementary Fig. 8). When the activity of the SGN terminals associated with the same IHC was averaged (see Methods) the frequency of Ca²⁺ transients was 1.62 ± 1.20 transients/ min (557 IHCs, 20 mice, P4-P9, Fig. 6d; Supplementary Fig. 1j). The activation of postsynaptic SGN terminals depends on the release of glutamate from IHC synaptic vesicles, which is a Ca²⁺ dependent process requiring the Ca²⁺ sensor otoferlin[58]. Therefore, we tested whether the absence of otoferlin abolished SGN Ca²⁺ signals in the cochlea of live mice. Since control and *Otof* knockout mice (*Otof*⁻/⁻) do not constitutively express GCaMP, we injected AAV-syn-jGCaMP8f directly into the cochlea of newborn mice in vivo. We found that Ca²⁺ signals were completely abolished in the large majority of SGN terminals in *Otof*⁻/⁻ compared to both wild-type (P < 0.0001, Dunn's tests, Kruskal-Wallis) or heterozygous (P < 0.0001) mice (Fig. 6e; Supplementary Fig. 1k).

To compare the Ca²⁺ activity of SGN terminals recorded in vivo and ex vivo, we investigated Ca²⁺ dynamics in SGN terminals from cochlear explants of *Snap25-GCaMP6s* or *GCaMP6f^{fl/fl}NeuroD1-Cre^{+/-}* mice at P7-P9, which is a time when the EP and the different ionic

composition of the endolymph and perilymph become established in the cochlea. We observed Ca²⁺ signals in the SGN terminals around some of the IHCs in ex vivo preparations, although their dynamics appeared both qualitatively and quantitatively different to that recorded in vivo conditions (Fig. 6f, Supplementary Fig. 9). The full duration at half maximum of the Ca²⁺ transients was significantly different than that recorded in age-matched SGNs terminals from live mice due to the presence of larger Ca²⁺ events (Fig. 6g). Similar to the ex vivo Ca²⁺ wave recordings (Fig. 5j–o), these longer Ca²⁺ events were not seen in vivo (Fig. 6g). The altered Ca²⁺ dynamics observed in cochlear explants resulted in the almost complete absence of correlated Ca²⁺ activity among SGN terminals both around individual and across several IHCs (P < 0.0001, Mann-Whitney U-test, Fig. 6h–j; Supplementary Fig. 1l).

When the in vivo Ca²⁺ activity of pillar and modiolar SGN terminals was investigated separately (Fig. 7a), we found that the frequency of spontaneous Ca²⁺ transients was generally higher on the pillar side of the IHCs compared to the modiolar side (P < 0.0001, Wilcoxon signed rank test, Fig. 7b). The average size of the response in active SGN terminals was also larger in the pillar compared to the modiolar SGN terminals (P < 0.0001, Wilcoxon signed rank test, Fig. 7c). Although the limitation imposed by Ca²⁺ imaging does not allow the resolution and quantification of individual synaptic events, these results indicate that some of the properties present in adult SGNs, with sensitive, high-spontaneous rate fibres situated on the pillar side of the IHCs and less sensitive lower-spontaneous rate fibres on the modiolar side[52,57,59,60],

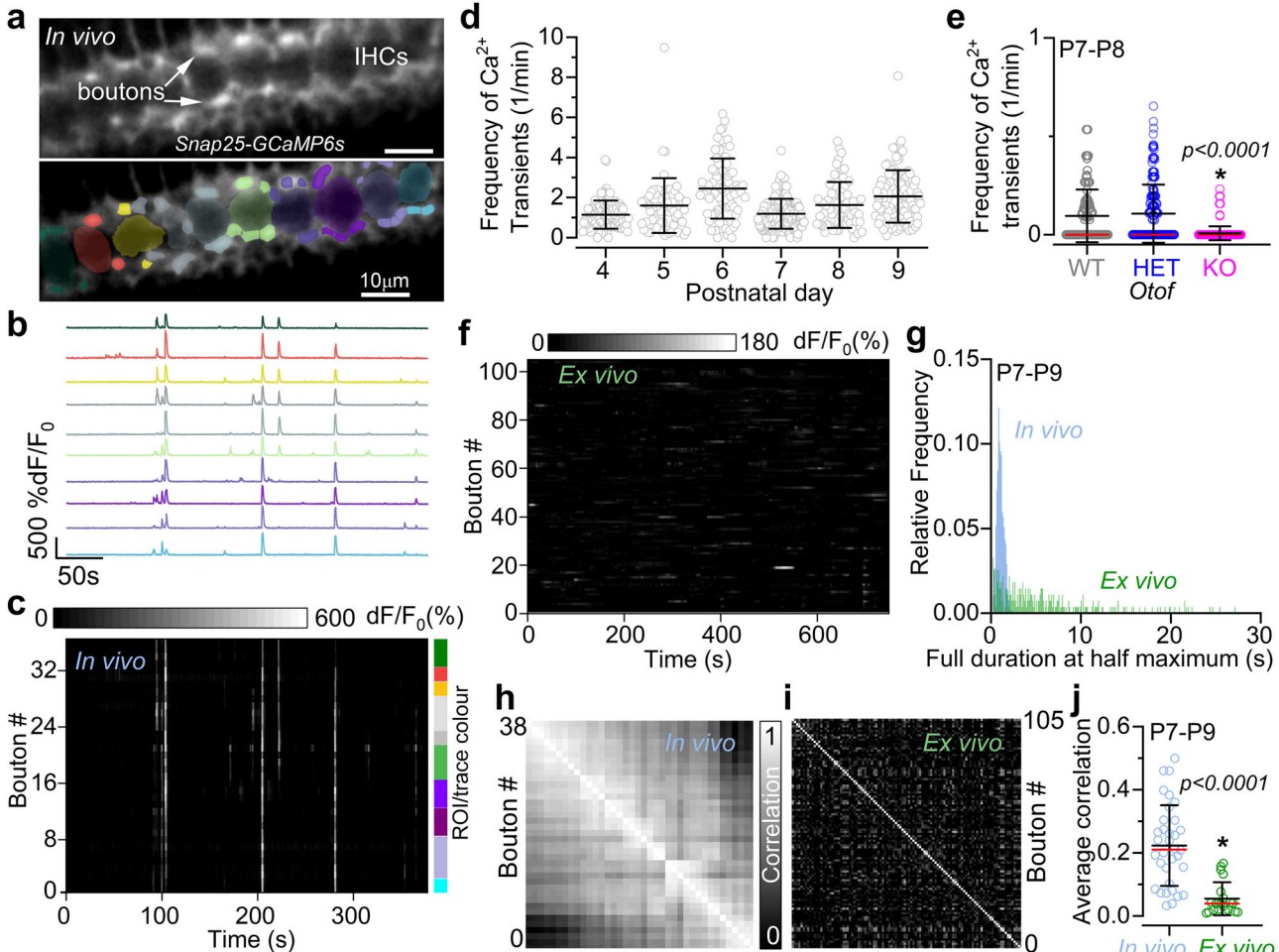

**Fig. 6 | Spontaneous activity in postsynaptic afferent terminals. a** Average intensity projection displaying GCaMP6 expression in vivo from a P4 mouse (top: *Snap25-GCaMP6s*) and segmentation mask highlighting ROIs (bottom) for identified SGN terminals colour-matched to their associated IHC. Fluorescence signals from individual ROIs in (**a**) shown as average traces per IHC (**b**) and individually (**c**: rasterplot, 38 SGN terminals). Colour labels on the right of the rasterplot indicate the SGN terminals belonging to the colour-matched IHC in (**a**) and (**b**). **d** Average Ca²⁺ transient frequency in SGN terminals per IHC as a function of age (IHCs from left to right: 97, 63, 76, 149, 67, 105, 20 mice, P4-P9). **e** Frequency of spontaneous Ca²⁺ transients in SGNs from wild-type (WT, grey), heterozygous (HET, blue) and knockout (KO, magenta) P7-P8 otoferlin (*Otof*) mice transduced in vivo with AAV-syn-jGCaMP8f. The frequency of Ca²⁺ transients is about 1/10th of that measured in mice that constitutively express GCaMP6 (**a**), which is due to the lower expression of GCaMP, reducing signal-to-noise and hampering the detection of smaller events. Calcium signals were present in SGN terminals from 57 wild-type IHCs (out of 125, 5 mice), 91 heterozygous IHCs (out of 203, 5 mice), but only in 9 *Otof⁻/⁻* IHCs (out of 214, 6 mice). **f** Calcium activity of individual ROIs placed on 105 SGN terminals recorded ex vivo (P8 *GCaMP6f^{fl/fl}NeuroD-Cre^{+/-}*). **g** Distribution of the full duration at half maximum of Ca²⁺ transients from P7-P9 SGN terminals recorded in vivo (blue, 1652 terminals, 11 mice) and ex vivo (green, 283 terminals, 11 mice). Correlation matrices computed from the in vivo (**h**) and ex vivo (**i**) recordings in panels (**c**) and (**f**), respectively. Each matrix element represents the Pearson correlation coefficient of one pair of SGN terminals. **j** Average correlation coefficient among SGN terminal Ca²⁺ signals recorded in vivo (34 recordings, 10 mice) and ex vivo (20 recordings, 11 mice) from P7-P9 mice expressing GCaMP. Average data: mean ± SD (median: red lines). Statistics: Kruskal-Wallis with Dunn's post-test (**e**); two-sided Mann-Whitney *U*-test (**j**).

are already present in the developing cochlea. When comparing the Ca²⁺ activity of SGN terminals assigned to the same IHC, we observed that not all burst of activity recruited all SGN terminals (Fig. 7d, Supplementary Movie 7). However, the fraction of the total number of SGN terminals activated around each IHC increased during synchronised Ca²⁺ transients across multiple IHCs ($P < 0.0001$, Mann Whitney *U*-test, Fig. 7e, Supplementary Fig. 8). A similar increase in synchronised Ca²⁺ transients across multiple IHCs was also observed when SGN terminals were segregated into modiolar and pillar ($P < 0.0001$ for both pillar and modiolar SGN terminals during single vs multiple IHCs responding, Dunn's post-test, Kruskal-Wallis, Fig. 7e). Moreover, the fraction of Ca²⁺ transients that recruited the entire pool of visible SGN terminals was larger during coordinated activity (22.1% of all events) compared to when a single IHC was active (1.4% of events). The increased recruitment of SGN terminals during IHC coordinated activity is most likely due to the stronger and long-lasting depolarisation of IHCs

compared to when recruited by the intrinsic spontaneous Ca²⁺ spikes. This is suggested by the evidence that the mean fluorescence amplitude of the Ca²⁺ transients in individual IHCs was significantly higher when 3 or more IHCs were simultaneously active (48 ± 38% dF/F₀, 1260 events) compared to when 1 IHC was showing spontaneous Ca²⁺ transients (25 ± 27% dF/F₀, 4288 events $P < 0.0001$, Mann Whitney *U*-test). Overall, these findings highlight the key role exerted by the synchronised activation of IHCs in providing a reliable and robust coordinated activation of the postsynaptic afferent terminals, which is required for the transmission of sensory-independent cues to the central auditory pathway.

## Discussion

Here we investigate spontaneous Ca²⁺ signal dynamics in the developing mammalian cochlea in vivo. We found that the primary sensory receptor IHCs intrinsically generate Ca²⁺ spikes, the frequency of which

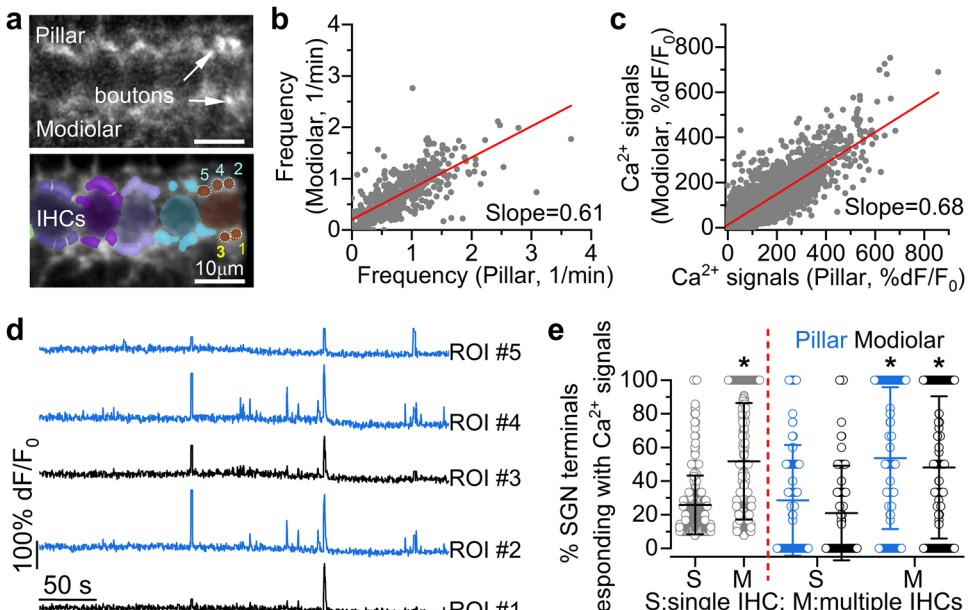

**Fig. 7 | Calcium transients in modiolar and pillar SGN terminals from the cochlea of live mice. a** Average intensity projection displaying GCaMP6 signal in vivo from a P4 mouse (top: *GCaMP6f^fl/fl^NeuroD-Cre^+/−^*) and ROIs highlighting individual SGN terminals assigned to the IHCs (bottom). **b** Scatterplot displaying the relationship between the frequency of Ca²⁺ transients in the SGN terminals positioned in the modiolar (2562) and pillar (2713) side of individual IHCs (495 IHCs, 20 mice). **c** Scatterplot displaying the relationship between the average amplitude of individual Ca²⁺ transients (3243) from SGN terminals positioned in the modiolar

and pillar side of individual IHCs. The amplitude of the response was averaged between the modiolar/pillar SGN terminals belonging to the same IHC displaying a response. **d** Fluorescence traces of the pillar (blue) and modiolar (black) ROIs numbered 1-5 in panel **a**. **e** Percentage of total (pillar and modiolar), pillar or modiolar SGN terminals showing Ca²⁺ transients when single (2439 events) or multiple ( > 2, 7442 events) IHCs were active. Average data are shown as mean ± SD. Statistical test in panel **e** is from two-sided Mann-Whitney *U-test*.

is increased by nearby spontaneous Ca²⁺ waves from supporting cells. Calcium waves also synchronise the activity of a few IHCs, which then rapidly spread longitudinally recruiting up to about 20 IHCs via a mechanism that is independent from Ca²⁺ travelling through the adjacent inner phalangeal supporting cells. We have also demonstrated that the synchronised activity of IHCs is essential for increasing the fraction of active postsynaptic SGN terminals both around individual and across several IHCs. These findings highlight significant insights into the mechanisms regulating spontaneous activity in the developing mammalian cochlea, which were not previously identified from ex vivo studies using cochlear explants. This also emphasises the need for maintaining the complex anatomy, innervation and physiology of the mammalian cochlea to understand how spontaneous activity controls the transmission of sensory-independent cues to the central auditory pathway.

The ability of pre-hearing IHCs to elicit spontaneous Ca²⁺ spikes in cochlear explants has been under scrutiny for several years. While some ex vivo studies suggested that Ca²⁺ spikes are intrinsically generated by IHCs for about six days after birth[19,20,34], others have proposed that they are only triggered by spontaneous Ca²⁺ waves originating in supporting cells[27,28]. These discrepancies were primarily driven by differences in experimental conditions used to maintain cochlear explants. We now show that in live mice, IHCs are intrinsically active throughout pre-hearing stages of development, which is primarily due to the establishment of the endocochlear potential (EP) during the second postnatal week in mice[33]. The EP drives a depolarising MET current into the IHCs, keeping their resting membrane potential close to the activation voltage of Ca$_V$1.3 Ca²⁺ channels (about -65 mV)[34,61]. Furthermore, the dynamics of the Ca²⁺ transients in developing IHCs was altered when using cochlear explants, since they become significantly less frequent.

Similar to ex vivo work[27,28,42], we found that the Ca²⁺ activity of several IHCs could be synchronised by regenerative intercellular Ca²⁺

waves originating in the supporting cells. The release of ATP from supporting cells causes the activation of purinergic auto-receptors leading to the opening of TMEM16A Ca²⁺-activated Cl⁻ channels and the efflux of K⁺, which depolarises nearby IHCs[28]. As seen for the IHC Ca²⁺ transients, Ca²⁺ wave dynamics were also largely altered in cochlear explants. In live mice, Ca²⁺ waves were faster and smaller in size compared to those reported from ex vivo recordings[8,36,37]. The very large Ca²⁺ waves reported from cochlear explants are most likely due to a response to cochlear damage since it has been shown that ATP can be released by supporting cells following noxious stimuli[49,50,62,63]. This was also supported by our in vivo recordings performed following the disruption of the cochlear physiological environment, which show large Ca²⁺ waves comparable to those recorded from cochlear explants. Interestingly, in our in vivo work we also found that the longitudinal propagation of activity amongst adjacent IHCs was about 3 times faster than that of the Ca²⁺ waves that triggered the synchronisation of IHC activity. Moreover, we could not see any Ca²⁺ increase in the inner phalangeal supporting cells in between the active IHCs, which have been implicated in the propagation of the Ca²⁺ signalling across IHCs[28]. This indicates that in vivo, the synchronised activation of multiple IHCs is initiated, but not driven by the slow ATP-induced Ca²⁺ waves previously described[8,27]. It is possible that the propagation of IHC depolarisation is caused by the increased K⁺ concentration in the intercellular space during their repolarisation, which is initiated by the few IHCs activated by the Ca²⁺ waves.

Understanding how the synchronous activity of the IHC is generated and regulated, for example by the inhibitory efferent cholinergic system, is crucial to identify how sensory-independent activity of the developing cochlea is able to influence the maturation of the central auditory pathway[10–13,64]. However, how developmental cues are faithfully transferred from IHC ribbon synapses to the SGN afferent terminals is still largely unknown[65,66]. Here we found that SGN terminals exhibit sporadic Ca²⁺ transients, which were observed both in vivo

and in cochlear explants. However, we also found that in live mice the periodic synchronisation of IHC activity was required to significantly increase the fraction of activated afferent terminals around each cell and across several IHCs. This periodic increase in the $Ca^{2+}$ signals is essential to drive the bursts of action potentials seen in the developing auditory fibres[7,14], which are expected to reinforce and refine neuronal connections to discrete areas of the ascending auditory pathway[10,13,67,68]. In the mature cochlea, SGN afferents are known to segregate around the IHC synaptic region, with their pillar side (towards the outer hair cells) contacted by low-threshold and high spontaneous-rate fibres, whereas high-threshold and low spontaneous-rate fibres primarily contact their modiolar side (towards the central axis of the cochlea)[57,69]. Although the molecular identity of the SGNs is primarily defined during pre-hearing stages of development[57,70–73], it is unknown when their pillar and modiolar functional segregation is initiated. We found that both the frequency and size of $Ca^{2+}$ transients were higher in the SGN terminals located on the pillar side of the IHCs compared to the modiolar side in live P4-P9 mice, indicating an onset of functional specialisation already during pre-hearing ages. The segregation of the SGN characteristics as early as P4 may suggest that it could be genetically encoded rather than determined by the spontaneous IHC activity.

In conclusion, our study has identified cellular mechanisms regulating the dynamics and nature of spontaneous activity in the pre-hearing mammalian cochlea in vivo. These findings, together with the molecular mechanisms identified using ex vivo work[19,27,28], will help to further our understanding on how the cochlea influences the refinement of the developing auditory system. This is also facilitated by the continuous development of genetically-encoded indicators, which when paired with our in vivo approach and that developed for the adult cochlea[74], will allow the investigation of auditory processing at single-synapse resolution in live mice.

## Methods

### Ethical statement
The animal work was licensed by the UK Home Office under the Animals (Scientific Procedures) Act 1986 (PPL_PCC8E5E93) and was approved by the University of Sheffield Ethical Review Committee (180626_Mar). Mice were kept in dedicated rooms with a 12 h light/dark cycle and both humidity and temperature were continuously monitored. For in vivo experiments (endocochlear potentials and imaging), mice were anaesthetised with isoflurane (2.0-2.5%) under oxygenation (0.8%) and killed by cervical dislocation at the end of the recordings (non-recovery procedure). For cochlear explant experiments, mice were killed by cervical dislocation. For in vivo AAV delivery in newborn pups (P1-P2), mice were anesthetised with isoflurane (2.5%) under oxygenation (0.8%). The body temperature of mice under anaesthesia was maintained using a heating mat (Harvard Apparatus). During the recovery from anaesthesia, mice were returned to their cage, placed on a heating mat, and monitored over the following 2–4 h.

### Animal strains and surgical settings for in vivo imaging
Experiments were performed using *GCaMP6f* floxed mice (*GCaMP6f^{fl/fl}*) from Jackson Lab. (stock number 028865). To drive the expression of *GCaMP6f* in different cell types within the cochlear sensory epithelium, we used the following Cre-lines. Hair cells were primarily targeted using *Atoh1-Cre* (Jackson Lab., stock number 011104) and *Myo15-Cre* (donated by Dr Safieddine)[75]. In these mouse lines, cre-dependent recombination, which is driven by the hair cell specific *Atoh1* or *Myo15* promoters, occurs during embryonic or early postnatal days, respectively. For IHCs, recordings were performed between P4 and P14 using *GCaMP6f^{fl/fl}Myo15-Cre^{+/-}* mice. Since in *GCaMP6f^{fl/fl}Myo15-Cre^{+/-}* mice cre-dependent recombination, which is driven by the hair cell specific Myo15 promoter, occurs from about P3 in the apical coil of the cochlea[75], we also used *GCaMP6f^{fl/fl}Atoh1-Cre^{+/-}* mice between P3 and P6.

The age-range overlap (P4-P6) allowed us to directly compare the recordings between the two mouse lines, which produced identical results. The constitutive *Snap25-GCaMP6s* mice (Jackson Lab., stock number 025111) and the *NeuroD1-Cre^{+/-}* mice (donated by Dr. Pavlínková)[76] were used to drive the expression of GCaMP6 in afferent fibres and terminals throughout the postnatal stages tested (P4-P9). *Pax2-Cre* mice (donated by Dr. Grove)[77] allowed the investigation of $Ca^{2+}$ signals in supporting cells and, to some extent, the hair cells throughout the postnatal stages tested (P3-P14). *Ca_V1.3^{-/-}* mice (P8-P10) were donated by Dr Striessnig[35] and *Otof^{-/-}* mice (P7-P8) by Drs Petit and Safieddine[58]. To image the cells in the cochlea in vivo, the above anaesthetised mice underwent a non-recovery surgical procedure as described in Fig. 1a-d. After removing the bone covering the apical coil of the cochlea (Fig. 1d), the space above was filled with extracellular solution heated at 37 °C composed of (in mM): 135 NaCl, 5.8 KCl, 1.3 $CaCl_2$, 0.9 $MgCl_2$, 0.7 $NaH2PO4$, 5.6 D-glucose and 10 Hepes-NaOH.

### Tissue preparation for ex vivo imaging
Inner hair cells (IHCs), supporting cells (SCs) and spiral ganglion neuron (SGN) terminals were studied in acutely dissected organs of Corti from pre-hearing mice. Organs of Corti were dissected using a solution containing (in mM): 135 NaCl, 5.8 KCl, 1.3 $CaCl_2$, 0.9 $MgCl_2$, 0.7 $NaH_2PO_4$, 5.6 D-glucose, 10 HEPES-NaOH. Sodium pyruvate (2 mM), amino acids and vitamins were added from concentrates (Thermo Fisher Scientific, UK). The pH was adjusted to 7.48 ( ~ 308 mmol kg$^{-1}$). The dissected apical coil of the cochlea, which corresponds to around the 12 kHz region[32], was transferred to a microscope chamber and immobilised with a nylon mesh fixed to a stainless-steel ring and viewed with the same two-photon imaging system used for the in vivo recordings (see below). Ex vivo imaging experiments were performed at room temperature to mimic the large majority of previous published studies[21,24,27,28,36,37,46,47].

### Endocochlear potential measurements
Borosilicate glass microelectrodes were filled with 150 mM KCl and mounted on a patch-pipette holder attached to a micromanipulator. Following the same procedure described for the in vivo surgery (Fig. 1a–d), the microelectrode was inserted into the scala media through the spiral ligament and the stria vascularis to measure the EP. The ground electrode was inserted in the lateral muscles of the neck region. EP responses were recorded under current-clamp mode (gap-free) using an Axopatch 200B amplifier (Molecular Probe, USA). Data acquisition was controlled using pClamp 10 software and a Digidata 1440 A (Molecular Devices, USA). Recordings were low-pass filtered at 1.0 kHz (8-pole Bessel), sampled at 10 kHz and stored on a computer for off-line analysis (Clampfit, Molecular Devices).

### AAV gene delivery in vivo
The surgical procedure for the delivery of AAVs was performed under anaesthesia (see: *Ethical statement*). For IHC transduction we used AAV-PHP.eB-jGCaMP8m with CMV promoter (custom made, Vector Builder); for SGNs we used AAV-syn-jGCaMP8f (#162376-AAV9, Addgene). The left ear was accessed via an incision just below the pinna[78]. When the round window membrane (RWM) was identified, it was gently punctured with a borosilicate pipette. This was followed by the injection of 1-2 µl of the AAV into the cochlea, which was the maximal titer achievable from the supplied AAV. Following the injection, the pipette was retracted from the RWM and the wound was closed with veterinary glue.

### Two-photon imaging
Following the surgical procedure, the anaesthetised mouse was transferred to the stage of a two-photon laser-scanning microscope (Bergamo II System B232, Thorlabs Inc., USA), equipped with a mode-locked laser system operating at 925 nm, 80 MHz pulse repetition rate

and <100-fs pulse width (Mai Tai, Spectra-Physics, USA). Images were acquired with a field size of 1024 pixel width and variable pixel height. The magnification of the microscope was adjusted for each experiment to contain the maximal area at the same focal plane in the field of view. The cochlea was localised and centred through the eyepieces using a low magnification objective. This lens was then switched to a higher magnification, water immersion objective (CFI75 LWD 16X W, NA 0.8; CFI75 Apochromat 25XC W, NA 1.1, Nikon), which was slowly lowered towards the surgical opening filled with extracellular solution. The cochlea was localised through the imaging software (Thorlabs Inc., USA) using landmarks produced by the autofluorescence of the surrounding tissue and opening in the bone. Imaging recordings were normally performed for up to 1 h before the mouse was killed by a schedule 1 method.

Comparisons between in vivo and ex vivo experiments were performed using the same mouse lines, imaging setup and acquisition settings. For both experimental conditions, the focal plane was set at the level of the IHC nuclear region.

## Image analysis

Images recorded with the two-photon system were saved on an external large-capacity storage for off-line processing. The image analysis consisted of the following steps: 1) Recordings were first examined visually using ImageJ (NIH) to identify and remove any frame interval that was affected by excessive drifts preventing signal detection. 2) Recordings were then filtered with a three-dimensional gaussian filter (2x2x2 pixels) to remove noise and improve signal detection. 3) Using custom python routines and GUI, intervals of frames during which the preparation was out of focus (e.g., due to breathing of the mouse) were identified and removed from the recording. This was performed by calculating a pixel-average of the fluorescence signal across a large region of interest (ROI). Sudden changes in focus usually appeared as either negative or (more rarely) positive transients in this trace. A combination of three methods was used to find these transients and remove them prior to data analysis: a) finding the (positive or negative) peaks of the transients using a prominence-based algorithm (*find_peaks* function of the *scipy* python module); b) template search throughout the trace and c) manual selection of the intervals not automatically identified. The result of the three methods was continuously supervised by the experimenter. Frames marked for removal were substituted by the last available in-focus frame to preserve the timing of the recording. In the fluorescence traces, these "missing" timepoints were substituted by a linear interpolation of the nearest fluorescence values. 4) After removing out of focus frames, each movie was adjusted for lateral drift using the NoRMCorre algorithm from the CaImAn package[79]. The pixel values of the motion-corrected movies were adjusted to ensure that the average was the same as the original recording.

For each type of experiment, a semi-automatic procedure was devised to segment the field of view in regions of interest and extract the fluorescence signal. Movies were opened in the Napari image software (10.5281/zenodo.3555620) equipped with custom plugins. ROIs and annotations were generated as "label" layers in Napari.

For the investigation of the IHC activity using either *GCaMP6f^fl/fl Myo15-Cre^+/-* or *GCaMP6f^fl/fl Atoh-Cre^+/-* mice, we first created an average image of the entire image stack. This was then fed to a Cellpose[80] algorithm (*cyto* model) to automatically generate a mask of ROIs for bright objects in the field of view. The masks were eroded by 1 to 3 pixels to minimise signal contamination between adjacent cells, and manually adjusted by the experimenter when required (e.g., to split merged cells or to join split ROIs). The masks of subsequent recordings of the same field of view were examined to manually ensure that the same IHCs were tagged with a unique label across recordings. This procedure allowed stitching fluorescence traces across several recordings. Different recordings were aligned using landmarks present in the field of view, such as bright supporting cells or a "double" row of

IHCs. The ROI mask was duplicated, and individual ROIs were annotated using a semiautomatic process to distinguish IHCs from other GCaMP-positive cells. Fluorescence traces for each ROI were extracted by calculating the average intensity of the pixels comprising a ROI for each frame, and the relative change of fluorescence intensity compared to baseline (dF/F$_0$) was used for further analysis.

For the analysis of supporting cell activity (*GCaMP6f^fl/fl Pax2-Cre^+/-* mice), we first drew a polygonal line across the location of the IHCs using an average image of the entire stack as reference. This allowed the calculation of the position and orientation of the Ca$^{2+}$ waves compared to the IHC position, and their longitudinal and radial extensions. Each image of the stack was then binned by a factor of 2 and manually thresholded. We used the Voronoi-Otsu labelling algorithm to semi-automatically generate three-dimensional ROIs that represented a Ca$^{2+}$ wave across different frames. These 3D ROIs were manually revised in Napari to merge and split events and remove artefacts. Events that involved Ca$^{2+}$ transients in individual isolated cells were excluded from the analysis. Recordings were then manually revised and bidimensional (2D) ROIs manually drawn across the maximal extension of Ca$^{2+}$ waves that were not selected by the preceding automatic method, or for which automatic detection was not satisfactory. These Ca$^{2+}$ waves were included in the calculation of the frequency and maximal extension of spontaneous events, but excluded from the calculation of their dynamical properties (e.g., expansion speed). Repetitive imaging recordings of the same field of view were collated and considered as one recording.

To compare Ca$^{2+}$ wave extension with the spread of IHC activity, we selected a subset of recordings where IHCs were visually distinguishable from the surrounding GCaMP6f-labelled supporting cells. Two kymographs (distance-time images) were generated by drawing a polygonal line across the location of the IHCs (IHC kymograph) and one 10 to 20 μm in front of the IHCs in the modiolar direction (GER kymograph). The two kymographs were superimposed and events that displayed activity in both the GER and IHC kymographs were labelled with two distinct ROIs using the "shape" layer of Napari. We ensured that ROIs representing the same event in the two kymographs superimposed in the temporal direction, so they could be directly compared in the following analysis. To calculate the speed of propagation of the Ca$^{2+}$ waves in the longitudinal direction, a line was drawn on the two kymographs from the origin of the wave up to the point of maximal expansion, and the speed computed as the ratio between the horizontal (space) and vertical (time) coordinates of the line relative to its origin.

For SGN synaptic terminal activity (*GCaMP6f^fl/fl NeuroD1-Cre* and *Snap25-GCaMP6s* mice), images were averaged, filtered with a white tophat filter with 10-20 pixels radius, and then segmented using the Voronoi-Otsu labelling algorithm. The labels produced by this algorithm were visually inspected, and ROIs manually adjusted using Napari. A custom-trained Cellpose model was used to semi-automatically label IHC bodies in the average image. Based on their proximity with the identified hair cells, SGN terminals were "assigned" to a specific IHC. For SGN terminals that were "in between" hair cells or for which an IHC could not be unequivocally assigned, we compared their activities with the two closest IHCs by calculating the correlation coefficient between the fluorescence traces, and automatically assigned the SGN terminals to the IHC displaying the highest correlation. SGN terminal labels were further annotated as modiolar, pillar or 'middle', depending on their position around the IHC body. All labels were inspected and manually adjusted. Fluorescence traces for each ROI were extracted by calculating the average intensity of its pixels for each frame, and the relative change of fluorescence intensity compared to baseline (dF/F$_0$) was used for further analysis.

## Calcium event detection

A custom GUI was created to inspect the traces and select "peaks" corresponding to Ca$^{2+}$ transients in individual IHCs or SGN terminals.

Peaks were first selected based on a prominence criterion (*find_peaks* function of *scipy*). The resulting identification was manually revised to correct artefacts and add missing events. Only peaks that had an amplitude larger than two times the standard deviation of a trace and a full duration at half maximum longer than 200 ms were considered as genuine. For IHCs, when the same field of view was recorded multiple times, we combined the peaks identified in subsequent recordings. Frequencies were computed as the total number of peaks divided by the total duration of the combined recordings. An "event" was defined as a $Ca^{2+}$ transient in one or multiple IHCs in the field of view. IHCs in the field of view were assigned to the same event if they had a peak within 2 seconds of one another. We considered "single" events that involved only one IHC and "multiple" events that involved three IHCs or more. A "multiple" event that involved IHCs with a gap of more than 4 not participating cells was considered as two separate events. This value was determined by observation of fluorescence movies.

Pearson correlation coefficients were calculated between the traces of every pair of IHCs in the field of view, provided that the two cells had at least five minutes of simultaneous recording (this was not always the case for repeated recordings of the same field of view where some cells were not consistently in focus). Before calculating the correlation coefficient, traces were detrended by subtracting a rolling median of the trace. Averages and standard deviations of correlation coefficients $r_s$ were calculated using the Fisher's z-transformation: $z = \mathrm{arctanh}(r_s)$; $\mathrm{avg}(r_s) = \tanh(\mathrm{avg}(z))$; $\mathrm{SD}(r_s) = \tanh(\mathrm{SD}(z))$.

To provide a semi-quantitative estimate of the firing rate of the action potential activity underlying the spontaneous $Ca^{2+}$ signals recorded using fluorescence microscopy, we used the "Global EXC" model of the CASCADE deep learning toolbox[40]. This model was trained on a diverse dataset of $Ca^{2+}$ and electrical spiking activity from excitatory neurons from different brain regions using several indicators, including GCaMP6f. The model output (spike probability) was converted into a firing rate by multiplying it by the framerate of the recording. The average firing rate was calculated as the cumulative sum of the spike probability divided by the total duration of the recording.

Full duration at half maximum of $Ca^{2+}$ waves was calculated by extracting a pixel-average trace of the $Ca^{2+}$ wave around its maximum. The frequency of $Ca^{2+}$ waves was calculated as the number of events divided by the total duration of the recording.

For SGN terminals, an "IHC" trace was generated by averaging the traces of all its associated SGN terminals and combining all the associated identified peaks (peaks in different SGN terminals within 1 second were considered the same hair cell peak). For SGN terminals, we identified as an "event" a $Ca^{2+}$ transient in one or multiple SGN terminals associated with an IHC. SGN terminals contacting the same IHC were assigned to the same event if they had a peak within 1 s of one another.

Frequency quantification and cross-correlation analysis were performed for recordings that lasted at least 5 min. For quantification of other properties of $Ca^{2+}$ signals, also shorter recordings were included in the analysis.

### Statistical analysis

Statistical comparisons were made by Student's *t*-test, Mann–Whitney *U*-test or Wilcoxon signed-rank test (when normal distribution could not be assumed). For multiple comparisons, analysis of variance (one-way ANOVA followed by a suitable post-test) was used for normally distributed data, otherwise Kruskal Wallis with Dunn's post-test was used. $P < 0.05$ was selected as the criterion for statistical significance. Only mean values with a similar variance between groups were compared. Average values are quoted in text and figures as means±S.D. Animals of either sex were randomly assigned to the different experimental groups. No statistical methods were used to define

sample size, which was defined based on previous published similar work from our laboratory. Animals were taken from several cages and breeding pairs over a period of several months.

### Reporting summary

Further information on research design is available in the Nature Portfolio Reporting Summary linked to this article.

## Data availability

Due to the large size, data generated and analysed in this study are available from the authors upon request. Source data are provided with this paper.

## Code availability

Code used in these analyses is available in the GitHub repository: github.com/Marcotti-Lab/invivo-IHC-cochlea.

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

## Acknowledgements

The authors thank Fabián Galindo Ramírez (Benemérita Universidad Autónoma de Puebla, Mexico) for helping on setting up the surgical procedure. Michelle Bird (University of Sheffield) for her assistance with the mouse husbandry. Niovi Voulgari and Catherine Gennery for their genotyping work and Matt Hool, Jing-Yi Jeng and Alice Zanella for the in vivo delivery of AAVs (University of Sheffield). We would also like to thank Stuart Johnson (University of Sheffield) and Corne Kros and Guy Richardson (University of Sussex) for their comments on an earlier version of the manuscript. This work was supported by the Wellcome Trust (224326/Z/21/Z) to WM. F.C. was supported by BBSRC (BB/V006681/1) to FC and WM. For the purpose of Open Access, the author has applied a CC BY public copyright licence to any Author Accepted Manuscript version arising from this submission.

## Author contributions

F.D.F., F.C. and W.M. collected and analysed the data and wrote the manuscript. W.M. and F.C. conceived and coordinated the study.

## Competing interests

The authors declare no competing interests.
