## [Peer review File · Nature Communications]

REVIEWER COMMENTS

Reviewer #1 (Remarks to the Author):

In this manuscript, De Faveri et al. demonstrate the influence of calcium dynamics within the cochlear neuroepithelium of pre-hearing mice on the activity of inner hair cells (IHCs) and their afferent synapses. Employing a surgical technique, they achieve optical access to the intact cochlear sensory epithelium in live mice, enabling single-cell resolution investigations into the relationship between activity in the GER and IHCs.

The noteworthy aspect of their work lies in the execution of technically demanding experiments that significantly contribute to our comprehension of the cellular mechanisms governing spontaneous activity in the pre-hearing Organ of Corti in vivo.

However, several considerations hinder a comprehensive assessment of their findings:

- The absence of a control group raises questions regarding the comparability of measurements. Could similar assessments be conducted in mice with damaged hair cells or in adult mice?
- The stability of the recorded signal remains unclear, particularly regarding the duration of recordings and potential variations between individual mice at the same postnatal stage. In general the authors should discuss the variation of activity in individual mice in the different “sessions”.
- A lack of information on the statistical tests employed alongside reported p-values impedes the evaluation of statistical comparisons.
- Discrepancies in the number of imaged cells across different postnatal weeks may influence statistical outcomes.
- The absence of a discussion section limits the interpretation and contextualization of results.
- Inadequate detail regarding the number and gender distribution of mice used for each experiment impedes reproducibility and thorough evaluation.

While the manuscript is very well written, certain technical aspects warrant clarification:

- Rationale behind the selection of the 8-18 kHz cochlear region and verification of its targeting. If not possible, maybe a change in the phrasing would be necessary.
- In Fig.3e, postnatal day 8 seems to be different than the others; why would that be?

- In Fig.4d, there seemed to be an increase until day P6 and then a drop in P8. Why is the author's interpretation? Was P7 measured?
- Potential impacts of anesthesia. Isoflurane is known to affect auditory brainstem responses, for example.
- Clarification on the interpolation method for missing frames and its uniformity across imaged time points is necessary.
- The authors mention that "In these mouse lines, cre-dependent recombination, which is driven by the hair cell specific Atoh1 or Myo15 promoters, occurs during embryonic or early postnatal days, respectively". – How would this affect the signal you are measuring during early postnatal days?
- Justification for the choice of extracellular solution filling the space above the cochlea.
- Why use 925 nm wavelength if you are imaging GCaMP6f?

Reviewer #2 (Remarks to the Author):

Thank you for the opportunity to review this work. This manuscript describes calcium transient activity in inner hair cells (IHCs) and their synaptic boutons, as well as supporting cells (SCs) of the greater epithelial ridge (GER) in the immediate pre-hearing cochlea. Most importantly, these recordings were performed in vivo through a cochlear window using mice expressing a genetically-encoded cytosolic calcium indicator (GCaMP), a major technical achievement. The authors have been pioneers in Ca imaging in the cochlea and have made many of the most significant contributions to understanding the role of Ca dynamics as a means of sensory-independent development and refinement of the cochlea, and this represents a major technical advance.

The experimental findings are presented clearly and thoroughly. However, I have some significant questions about the methodology and conclusions that should be addressed to allow better understanding of the impact of this work. I agree that the performance of truly live, in vivo imaging performed in this study is inherently impactful to a certain degree. However, the actual impact on our understanding of the mechanisms and roles of ICS activity in the developing cochlea (versus simply a major technical advance) hinges on two questions:

1. Is the in vivo technique meaningfully different (and better) than the ex vivo technique?

2. Do the observations made in this system tell us things we didn't already know about these mechanisms, which have been studied extensively cochlear explants. That is, what are the actual scientific advances from this study?

Question #1: Comparison of the in vivo and ex vivo techniques.

The authors talk about maintenance of endocochlear potential, separation of compartments, and efferent feedback as specific advantages. However, EP in cochlea develops slowly over the first 2 postnatal weeks; over the recording period in this paper (P4-P10), EP magnitude is only roughly 10-30% of adult maximum. Is the EP difference between the in vivo and ex vivo conditions actually important to ICS activity? Furthermore, the impact of efferent feedback on the Ca activity being studied is unclear. The ionic environment is more obviously connected, but that is not directly studied or addressed here.

I ask that the authors provide a clearer explanation and demonstration of how their in vivo measurements compare to what has been found ex vivo. They do mention some comparisons (for example, the rate of IHC Ca transients is the same, but the width of IHC Ca transients is faster in vivo; and there are differences in the speed, size, and evolution of SC ICS waves). How do the authors explain these differences? There are a number of technical (for instance, imaging plane, which can affect measurement of speed and directionality of wave propagation when analyzed in 2D; imaging parameters, which can affect segmentation and ROI selection) as well as biological (effect of EP, ionic environment, efferent feedback) factors that may affect these differences. So for the reader to be able to interpret these differences and understand which are meaningful, there should at least be some attempt to explain why they could be happening.

I feel the manuscript would be improved by:

- Compiling these differences in a more organized manner
- Making some attempt to explain why these differences would occur; and
- Ideally, performing additional experiments could be performed to test some of these biological hypotheses. For instance, the EP could be measured in this preparation (to confirm that it is indeed being preserved, but also to see if changing the EP affects these Ca parameters). Or the ionic composition of the perilymph/endolymph could be altered. These are, of course, technically challenging, but would be necessary to connect the findings from this preparation to the large body of literature performed on ex vivo explants.
- More thorough discussion about the impact of the imaging technique (imaging plane, analysis) on the measured parameters.

Question #2: What are the scientific conclusions/discoveries from this work?

The authors make some observations about changes in Ca transient/wave behavior over the course of development. In particular, they find that, in the second week, IHC transients are longer and smaller, and SC ICS waves are longer. However, they analyze these changes in different ways, sometimes showing day-by-day data (figure 1k and 3e) and sometimes aggregating by 1st vs 2nd postnatal week (figures 2e and 3d). The dichotomization between 1st and 2nd weeks seems somewhat arbitrary, especially since they only perform recordings from P4-P10, and the variance in certain parameters (fig 3e) between individual days appears to be as large, if not greater, than the variance between "weeks" (fig 3d). I feel the manuscript would be improved with:

- more consistent and/or rigorous statistical analysis of time-based trends
- clearer discussion of why these time-based trends may be occurring, informed by previous abundant literature on the evolution of these phenomena over this time period.

The much more potentially significant finding from this work relates to the relationship between SC ICS waves and IHC transients. It is generally thought that IHC transients are coordinated directly as a result of SC ICS waves (through mechanisms involving extracellular ATP, K⁺ efflux, and depolarization of IHCs near these ATP release sites). However, it remains possible that the IHC Ca transients are coordinated (and have propagating waves) through mechanisms intrinsic to the IHCs themselves. In this study, the authors observe that the IHC waves propagate faster and farther than the SC waves. This implies that there is some IHC-intrinsic mechanism for their coordinated activity, and could represent a significant shift in how we understand how spontaneous, coordinated IHC activity occurs in late cochlear development. It is confusing, however, when they also conclude later that the "key role exerted by the Ca waves from SCs... in coordinated activation of the postsynaptic afferent terminals." The synaptic bouton activity was only correlated with respect to IHC transient activity, not adjacent SC activity.

Overall, this potentially significant finding feels underinvestigated in this study. For example, can the authors shed any light on why this is happening? Is it an IHC-intrinsic propagation? Or is this happening because of extracellular spread of some paracrine signal (ATP, K), and/or different thresholds for triggering a Ca increase in a particular cell type? There are many questions.

I feel the manuscript would be improved by:

- Additional analyses/experiments that describe more thoroughly what the relationship is between IHC and SC waves. Figures 3j-k show this in aggregate, it is unclear how frequently the IHC/SC waves co-occur overall. Also, in 3j/k: how many mice were tested? how many events/mouse? Why are there different numbers of events quantified in these two plots (3j: distance; 3k: speed)?
- Use of drugs or experimental manipulations to try to assess these connections more precisely would be difficult, but necessary to be able to make strong conclusions about the relationship between ICS waves and IHC transients.

- At the very least, in the absence of additional experiments, a thorough discussion of this observation and its relationship to prior findings is required.

Reviewer #3 (Remarks to the Author):

Summary:

De Faveri and colleagues developed and described a novel method that utilize high resolution Ca²⁺ multiphoton imaging enabling the visualizing of calcium signals from different cellular subtypes in the cochlea, from anesthetized prehearing mice in vivo. This newly developed approach keeps the cochlear integrity (although still partially invasive) to investigate the molecular mechanisms underlying cochlear development and refinement for the auditory circuitry, here focusing on the Ca²⁺ waves occurring spontaneously in the organ of Corti.

The authors use different mouse models to express the genetically encoded Ca²⁺ indicator GCaMP6 in different cell types including inner hair cells (IHCs), supporting cells of the greater epithelial ridge (GER) and the bouton endings of the type-I auditory nerve fibers (ANFs). They demonstrate that IHCs, GER cells and type-I ANF bouton terminals display spontaneous Ca²⁺ signals in a more or less coordinated fashion, whereby Ca²⁺ waves are spontaneous generated by supporting cells in the GER, synchronizing the activity of several IHCs that at the end depolarize the type-I ANFs.

The approach used here is derived from a method already described, by the laboratory of Anthony Ricci (Stanford University). Secondly, the main result showing postnatal calcium waves in the organ of Corti in different cell types and how this coordinates, has also been shown extensively before, in excised cochlear tissue, in vitro (Bergles lab, Marcotti lab, Mammano lab and Gale lab). However, showing the known phenomenon of postnatal cochlear calcium waves in live animals, provides a 'quantum jump' step forward, providing high assurance that the phenomenon is actually occurring and of high significance in the living animal. This approach therefore provides a major breakthrough and will allow to gain future new insights in molecular mechanisms that sustain cochlear developments, nerve fiber refinement as well as mechanism that could participate to setting the encoding properties of the ANFs (modiolar vs pillar).

Overall, this study is well designed, straightforward with solid results and properly discussed in respect to the available literature. Comments provided here are minor:

Minor comments:

1) Supplementary figures with some raw traces of Ca²⁺ imaging recordings from IHCs, SCs and ANFs before off-line image treatment (i.e. removing out-of-focus frames, frames due to animal breathing, ...) should be added, for the reader to understand what to expect from such raw data.

2) I am not questioning the validity of the Ca²⁺ imaging from ANFs but I am wondering if authors have observed different patterning of Ca²⁺ signals throughout the 2 weeks of development as during this period there is a lot of fiber pruning and refinement. Can authors discuss if they believe their approach could identify branching fibers vs individual fibers? Does the proportion of modiolar vs pillar fibers change over these 2 weeks?

3) p values, stars showing significance, and statistical tests are needed on graphs/figure legends.

4) Please provide more explicit details (in the method section) regarding animal age range for each mouse line.

5) Line 93: Suggest to also include Babola et al., 2020 and 2021 showing P2Ry1 expression in the supporting cells of the GER.

6) Line 96: Suggest to add the following reference regarding Ca²⁺ waves and cochlear tissue damage: Nowak et al., 2021: <https://doi.org/10.1523/ENEURO.0383-21.2021>

7) Supplementary movie 6 is not listed in the main text.

Reviewer #4 (Remarks to the Author):

In vivo spontaneous Ca²⁺ activity in the pre-hearing mammalian cochlea

We are very grateful to all three Reviewers for the several insightful and constructive comments, which have helped to strengthen the revised manuscript. We believe that we have addressed all key points raised by the Reviewers by performing several additional experiments and further analysis. Line numbers listed in our reply below refer to the Revised pdf manuscript file with “Changes Highlighted”.

We would like to apologise for the “compact” submitted article, which was originally submitted for a short format publication for another Spring Nature journal and was then transferred to Nature Communications before being sent to Reviewers.

Summary of key aspects of our revision:

1) Control/validation: this is an excellent point that has required several new experiments. One key point to consider from our new *in vivo* approach is that we work with a completely sealed cochlear partition, which limits our ability to manipulate (e.g., pharmacologically) any physiological activity during an experiment. One alternative approach is to use transgenic or mutant mice targeting genes crucial for cochlear development and function. Therefore, we have addressed this point by using the two available knockout mouse lines, which specifically disrupt Ca²⁺ transients (*Cav1.3*^{-/-}) and synaptic transmission (*Otof*^{-/-}) in the developing inner hair cells (IHCs). These new experiments have demonstrated that Ca²⁺ signals in IHCs and spiral ganglion neurons (SGNs) were near or completely abolished when using the above knockout mice (see new **Fig. 2d** and **Fig. 6e**, respectively). Of course, there are many more available mouse models we could test, but this will require importing (and rederiving) them in our pathogen-free animal facility, establishing the new colonies, and then performing the experiment/analysis, which will require considerable time (i.e., several years).

To provide additional evidence for the reliability of our *in vivo* approach, we have also performed three additional experiments:

a) Recorded Ca²⁺ transients in IHCs and Ca²⁺ waves in the supporting cells at, and just after, the onset of hearing (P12 and P14). These new data have demonstrated that Ca²⁺ signals in both IHCs and supporting cells disappear with the onset of hearing (see new **Fig. 2a** and **Fig. 4e**), which is in line with our understanding of how cochlear physiology operates.

b) Recorded Ca²⁺ waves in the supporting cells following the perforation of the membrane sealing the cochlear partition to remove the physiological environment i.e. the separation of the cochlear fluids and EP (see new **Fig. 4h,i,l,m** and **Supplementary Figure 5**).

c) Recorded Ca²⁺ transients in IHCs and SGN terminals, and Ca²⁺ waves in the supporting cells from cochlear explants (*ex vivo*). This new data is shown in **Fig. 2e-g** (IHCs), **Fig. 4j,k,l,m** and **Supplementary Figure 5** (supporting cells) and **Fig. 6f-j** and **Supplementary Figure 8** (SGNs). We found that the Ca²⁺ dynamics under unphysiological recordings conditions (cochlear explants & *in vivo* but with an open cochlear partition) were comparable but largely altered compared to those recorded under *in vivo* physiological conditions.

2) Summary of the key difference between *in vivo* and *ex vivo* approaches: we agree that the several differences we identified between *in vivo* and cochlear explant approaches were somewhat hidden in the submitted text. In the much-extended version of revised manuscript, which now also includes a Discussion section on this point, we have provided a better description of the findings. Some of these differences were also identified following the several new experiments suggested by the three Reviewers (see point 1 above). Some of the key findings are briefly listed below:

a) The sensory IHCs exhibit spontaneous Ca²⁺ transients in the absence of any Ca²⁺ waves from the supporting cells, a conclusion that was not supported by several published studies using cochlear explants (e.g., Tritsch et al., 2007; Tritsch et al., 2010; Wang et al., 2015).

b) Although the synchronised activity of IHCs is initiated by the spontaneous Ca²⁺ waves originating in the supporting cells, its longitudinal spread occurs via a faster Ca²⁺-independent mechanism.

This is different from the currently proposed mechanism described in the explanted cochlea, which only implicates Ca^{2+} waves from supporting cells.

- c) The majority of spiral ganglion neuron (SGN) afferent terminals innervating individual IHCs only become simultaneously activated during the synchronised Ca^{2+} activity of several IHCs, again a new mechanism not identified in cochlear explants.
- d) The SGN afferent terminals exhibit functional segregation according to their position around individual IHCs (pillar vs modiolar), which is a key characteristic of adult neurons. The appearance of this important functional feature, which allows IHCs to encode a wide range of sound intensities, at pre-hearing stages of cochlear development was previously unknown.
- e) We also found that the key characteristics of Ca^{2+} signalling dynamics in IHCs, SGN terminals and supporting cells are significantly altered when experiments are not performed under *in vivo* physiological recording conditions. Crucially, unphysiological recordings (*ex vivo* or following the perforation of the membrane sealing the cochlear partition) leads to very large Ca^{2+} waves, which are likely to be driven by cochlear damage signals instead of functionally relevant responses.

We hope that the Reviewers will consider satisfactory our extensive effort in addressing all key points raised during the initial submission, especially considering the challenging nature of the proposed *in vivo* recordings and analysis.

Reviewer #1:

In this manuscript, De Faveri et al. demonstrate the influence of calcium dynamics within the cochlear neuroepithelium of pre-hearing mice on the activity of inner hair cells (IHCs) and their afferent synapses. Employing a surgical technique, they achieve optical access to the intact cochlear sensory epithelium in live mice, enabling single-cell resolution investigations into the relationship between activity in the GER and IHCs.

The noteworthy aspect of their work lies in the execution of technically demanding experiments that significantly contribute to our comprehension of the cellular mechanisms governing spontaneous activity in the pre-hearing Organ of Corti *in vivo*.

However, several considerations hinder a comprehensive assessment of their findings:

- The absence of a control group raises questions regarding the comparability of measurements. Could similar assessments be conducted in mice with damaged hair cells or in adult mice?

This is an excellent point which we believe we have addressed in **point 1** of the above “Summary of key aspects of our revision”. In brief, we have performed four sets of new experiments:

- i) Used two knockout mouse lines known to disrupt Ca^{2+} signals in the IHCs (*Cav1.3^{-/-}*: new **Fig. 2d**) and SGNs (*Otof^{-/-}*: new **Fig. 6e**).
- ii) performed recording of Ca^{2+} transients in IHCs and Ca^{2+} waves in the supporting cells between P12 and P14 (see new **Fig. 2a** and **Fig. 4e**), which is at and just after the onset of hearing in mice.
- iii) Recorded Ca^{2+} waves in the supporting cells following the perforation of the membrane sealing the cochlear partition to disrupt the physiological environment (new **Fig. 4h,i,l,m** and **Supplementary Figure 5**).
- iv) Recorded Ca^{2+} transients in IHCs and SGN terminals, and Ca^{2+} waves in the supporting cells from cochlear explants (*ex vivo*). This new data is shown in **Fig. 2f,g** (IHCs), **Fig. 4j,k,l,m** and **Supplementary Figure 5** (supporting cells) and **Fig. 6f-j** and **Supplementary Figure 8** (SGNs).

- The stability of the recorded signal remains unclear, particularly regarding the duration of recordings and potential variations between individual mice at the same postnatal stage. In general the authors should discuss the variation of activity in individual mice in the different “sessions”.

We now stated in the Methods section that imaging recordings were normally performed for up to 1 hour for ethical reasons, which is not stated in the revised manuscript (**ln.110-111**). We have also added one of these long recordings in the new **Supplementary Figure 1**. During these recordings, we did not observe any changes in activity or baseline fluorescence. Each mouse is only used once, so we cannot provide information about “*variation of activity in individual mice in the different sessions*”. We could not identify any specific data variability among different experimental sessions (within a day and across days). The observed data variability in the recordings was due to the use of

live mice, which we believe is quite standard for biological systems. Variability in data collection is also present in recordings from cochlear explants (this study and many more present in the literature), where several aspects of the experiments can be easily controlled across animals. We have plotted below the average frequency of Ca^{2+} signals in IHCs, SCs and SGN terminals for each experimental mouse for both the *in vivo* and *ex vivo* recordings. We hope that the above considerations address the point raised by the Reviewer.

- A lack of information on the statistical tests employed alongside reported p-values impedes the evaluation of statistical comparisons.

We thank the reviewer for this point since we realised that, despite mentioning the statistical tests in most of the data presented (either in the text or figure legend), some were missing. We have now fully addressed this point in the extended revision of the manuscript.

- Discrepancies in the number of imaged cells across different postnatal weeks may influence statistical outcomes.

Most of the statistical tests used in the submitted version were appropriate for the analysis of the different data sets. However, following a consultation with a statistician, we have re-analysed some of our data. Each set of data was tested for normal distribution prior to the selection of the statistical test. If normally distributed, we used t-test or ANOVA which are known to be robust even for unequal samples when the variance is comparable. When the data were not normally distributed, we used non-parametric tests, like the Mann-Whitney U test or the Kruskal-Wallis test, which are not affected by the shape of the distribution, even when sample sizes differ.

- The absence of a discussion section limits the interpretation and contextualization of results.

The Discussion is now included in the revised manuscript.

- Inadequate detail regarding the number and gender distribution of mice used for each experiment impedes reproducibility and thorough evaluation.

The number of mice used is now listed throughout the revised manuscript (either the Figures or Figure legends).

We also appreciate the point made by this Reviewer about gender distribution. However, we stated in the submitted manuscript that “*Animals of either sex were randomly assigned to the different experimental groups*”. This is for two main reasons: **a)** we and other groups have extensively investigated this aspect in a few recent publications (e.g. Jeng et al., 2020, *JPhysiol*, 598:3891-3910; Jeng et al., *JPhysiol*, 2020, 598:4339-4355; Dondzillo et al., 2021 *Hear Res*, 404:108215; Jeng et al., 2021, *JPhysiol*, 599:269-287). We found that the morphological and biophysical properties were not affected by sex until late adulthood, which is very far from the range used in our study (pre-hearing). Because of this large amount of previous work, we now combine the data from both sexes, which is the reason for writing the cited sentence in the Method. **b)** The experiments performed in this manuscript are extremely challenging and it would be unfeasible and unethical to double all experiments based on mouse gender, unless there is clear evidence supporting the gender differences, which we do not have (see point **a** above). We hope that the above explanations are satisfactory.

While the manuscript is very well written, certain technical aspects warrant clarification:

- Rationale behind the selection of the 8-18 kHz cochlear region and verification of its targeting. If not possible, maybe a change in the phrasing would be necessary.

The 8-18 kHz cochlear region is the one accessible under our experimental conditions and the position is calculated based on the frequency map of the mouse using Muller et al., (2005), which is now cited in the manuscript (ln. 85-87).

- In Fig.3e, postnatal day 8 seems to be different than the others; why would that be?

The two-way ANOVA post-test shows that P8 is only slightly significantly higher than P9 ($P = 0.0495$), most likely because the number of Ca^{2+} waves are starting to decrease progressively. We have now included the recordings at P12 and P14 (new Fig. 4e) and shown that Ca^{2+} waves no longer occur from the onset of hearing (see also Results: ln. 187-187).

- In Fig.4d, there seemed to be an increase until day P6 and then a drop in P8. Why is the author's interpretation? Was P7 measured?

We have now added the recordings at P7. We could not find any specific trend in the data between P4 and P9, as also indicated by the statistical analysis ($P = 0.6537$), which is now stated in the Figure legend.

- Potential impacts of anesthesia. Isoflurane is known to affect auditory brainstem responses, for example.

The Reviewer is correct in stating that isoflurane is known to affect the brain electrical activity. We attempted to address the issue directly by using an injectable anaesthetic such as a ketamine/xylazine (Ket/Xyl) mixture, which are widely used to assess auditory responses with ABRs. The problem with working with pups is that it is extremely difficult to induce and maintain deep anaesthesia with injectable anaesthetic, especially during long-lasting *in vivo* recordings. With the help of our VET at the University of Sheffield, we attempted these experiments with different combinations and doses of injectable drugs, including Ket/Xyl, without success. Pups were either dying within 30-45 minutes from the start of the surgery or did not reach sufficiently deep anaesthesia to support our invasive surgical procedure.

Although we could not use injectable anaesthetics, we believe we have at least in part addressed this point by performing the following experiments:

- The *in vivo* endocochlear potential (EP) depends on the biophysical/electrical characteristics of the different cell types within the cochlea, including the hair cells and supporting cells. As requested by Reviewer 2, we performed EP measurements and found that, under our experimental conditions (i.e., isoflurane), the values at P7 and P10 were matching those recorded in mice anaesthetised with Ket/Xyl (Li et al., 2020, Front Cell Neurosci 14, 584928). We now mention this in the text (ln. 95-99).
- We performed new *in vivo* experiments where the membrane sealing the cochlear partition was ruptured, leading to the mixing of the cochlear solutions and the disappearance of the EP. Under this unphysiological *in vivo* condition (see Fig. 4h,i,l,m and Supplementary Figure 5), in which mice are still under the effect of isoflurane, we found the dynamics of Ca^{2+} waves in the supporting cells mimicked those obtained from cochlear explants (in the absence of anaesthetic), indicating that isoflurane has no or negligible effect on the Ca^{2+} signals in the developing cochlea. We now mention this in the text (ln. 188-205).
- Finally, we have shown that the frequency of the firing rate extrapolated from the *in vivo* Ca^{2+} signals (under isoflurane anaesthesia) matched that recorded in early postnatal IHCs from cochlear explants using physiological cell-attached patch clamp (ln. 138-143). At this age (first postnatal week), the EP is not yet present in the cochlea making the *in vivo* and the *ex vivo* approach more comparable.

- Clarification on the interpolation method for missing frames and its uniformity across imaged time points is necessary.

The exact procedure is stated in the Method section (ln. 493-496): "Frames marked for removal were substituted by the last available in-focus frame to preserve the timing of the recording. In the

fluorescence traces, these “missing” timepoints were substituted by a linear interpolation of the nearest fluorescence values.”

- The authors mention that “In these mouse lines, cre-dependent recombination, which is driven by the hair cell specific *Atoh1* or *Myo15* promoters, occurs during embryonic or early postnatal days, respectively”. – How would this affect the signal you are measuring during early postnatal days?

We apologise for not having made this clear in the submitted manuscript. We have now added a few sentences in the Methods to explain this more clearly (ln. 409-425).

- Justification for the choice of extracellular solution filling the space above the cochlea. The solution used is a classical extracellular saline solution that is keeping all the tissues around the cochlea in a more physiological environment.

- Why use 925 nm wavelength if you are imaging GCaMP6f?

This is the best wavelength for 2 photon imaging of GCaMP6, being very close to the excitation maximum: see <https://www.janelia.org/lab/harris-lab/research/photophysics/two-photon-fluorescent-probes>

Reviewer #2:

Thank you for the opportunity to review this work. This manuscript describes calcium transient activity in inner hair cells (IHCs) and their synaptic boutons, as well as supporting cells (SCs) of the greater epithelial ridge (GER) in the immediate pre-hearing cochlea. Most importantly, these recordings were performed *in vivo* through a cochlear window using mice expressing a genetically-encoded cytosolic calcium indicator (GCaMP6), a major technical achievement. The authors have been pioneers in Ca imaging in the cochlea and have made many of the most significant contributions to understanding the role of Ca dynamics as a means of sensory-independent development and refinement of the cochlea, and this represents a major technical advance.

The experimental findings are presented clearly and thoroughly. However, I have some significant questions about the methodology and conclusions that should be addressed to allow better understanding of the impact of this work. I agree that the performance of truly live, *in vivo* imaging performed in this study is inherently impactful to a certain degree. However, the actual impact on our understanding of the mechanisms and roles of ICS activity in the developing cochlea (versus simply a major technical advance) hinges on two questions:

1. Is the *in vivo* technique meaningfully different (and better) than the *ex vivo* technique?

This is an excellent question which we believe we have addressed in **point 2** of the above “**Summary of key aspects of our revision**” (see Pg. 1). The several additional experiments performed during this revision have allowed us to provide a more direct comparison of the Ca²⁺ signal dynamics between the *in vivo* approach and that of unphysiological recording conditions (cochlear explants & *in vivo* but with an open cochlear partition). Not only are most of the key aspects of Ca²⁺ signalling in the developing cochlea different between the two approaches but, more importantly, we have identified new mechanisms in live mice that are not present in cochlear explants. All of these aspects are now highlighted in the extensively re-written Results section and Discussion.

Therefore, the short answer to the above question is yes, the *in vivo* approach provides a more realistic readout of the true mechanisms occurring in the developing cochlea.

2. Do the observations made in this system tell us things we didn't already know about these mechanisms, which have been studied extensively cochlear explants. That is, what are the actual scientific advances from this study?

As mentioned in the above question 1, we have discovered new mechanisms that are different or have not been identified in cochlear explants. Moreover, we now show that several mechanisms occurring in the cochlear explants are not present *in vivo* and, in some cases, such as the large Ca²⁺ waves recorded from cochlear explants, are most likely generated by damage signals occurring in the dissected cochlea.

Question #1: Comparison of the *in vivo* and *ex vivo* techniques.

The authors talk about maintenance of endocochlear potential, separation of compartments, and efferent feedback as specific advantages. However, EP in cochlea develops slowly over the first 2 postnatal weeks; over the recording period in this paper (P4-P10), EP magnitude is only roughly 10-30% of adult maximum. Is the EP difference between the *in vivo* and *ex vivo* conditions actually important to ICS activity? Furthermore, the impact of efferent feedback on the Ca activity being studied is unclear. The ionic environment is more obviously connected, but that is not directly studied or addressed here.

Regarding the importance of the EP for hair cell physiology: we have performed additional experiments to measure the EP under our imaging conditions, which matches that previously measured in the developing mouse cochlea using the anaesthetic mix ketamine/xylazine (ln. 95-99).

The establishment of the EP during the second postnatal week is tightly linked to the acquisition of the mature endolymphatic ionic composition that differs from that of the perilymph, especially in terms of Ca^{2+} concentration, which is key for mechano-electrical transduction. The importance of these changes is evident from the observation that IHCs generate spontaneous Ca^{2+} transients throughout pre-hearing stages *in vivo*, but not in cochlear explants, which instead require Ca^{2+} waves to be depolarised (e.g., Trisch et al 2007). This is because *in vivo*, the low Ca^{2+} concentration in the endolymph (about 0.3mM: Johnson et al., 2012 JNeurosci 32:10479) opens the MET channels leading to a depolarizing MET current into the IHCs driven by the EP.

Regarding the efferent feedback: this is a key difference between the *ex vivo* and *in vivo* recordings, being missing in the former experimental condition. As mentioned in the revised manuscript, we made a couple of observations regarding the potential role of the efferent systems (ln. 156-158 and ln. 217-221). However, we do not currently have the correct mouse models to investigate this aspect in detail. Although we are planning to import a couple of transgenic mice where efferent signalling is affected, this is going to take another few years.

Regarding the importance of the ionic environment for supporting cell function: as mentioned above, during the second postnatal week the Ca^{2+} concentration in the endolymph drops from about 1.3 mM (first postnatal week) to about 0.3 mM Ca^{2+} . This is important not only for the function of IHCs (see above), but also for the supporting cells since they express connexin 26 and 30 (Cx26 and Cx30) hemichannels, the activation of which is highly dependent on the extracellular Ca^{2+} concentration (e.g., Bayraktar et al., 2024 Int. J. Mol. Sci. 25:6594). Published studies using cochlear explants have used high Ca^{2+} concentrations (1.3-1.5 mM), which has been shown to decrease the open probability of connexin hemichannels expressed in supporting cells and decrease the release of ATP (e.g., Mammano, 2019 Cold Spring Harb Perspect Med 9), which is likely to contribute to the different Ca^{2+} dynamics observed in the Ca^{2+} waves between *ex vivo* and our *in vivo* work.

I ask that the authors provide a clearer explanation and demonstration of how their *in vivo* measurements compare to what has been found *ex vivo*. They do mention some comparisons (for example, the rate of IHC Ca transients is the same, but the width of IHC Ca transients is faster *in vivo*; and there are differences in the speed, size, and evolution of SC ICS waves). How do the authors explain these differences? There are a number of technical (for instance, imaging plane, which can affect measurement of speed and directionality of wave propagation when analyzed in 2D; imaging parameters, which can affect segmentation and ROI selection) as well as biological (effect of EP, ionic environment, efferent feedback) factors that may affect these differences. So for the reader to be able to interpret these differences and understand which are meaningful, there should at least be some attempt to explain why they could be happening.

We thank the Reviewer for asking us to perform a more direct comparisons between the *ex vivo* and the *in vivo* work. As highlighted in point 2 of the above “Summary of key aspects of our revision” (see Pgs. 1-2), we have performed a large number of additional experiments using either cochlear explants or *in vivo* but with an open cochlear partition, which are both unphysiological compared to our *in vivo* approach. Most of the key aspects of Ca^{2+} signal dynamics in the developing cochlea are different between the *in vivo* and the unphysiological approaches. Moreover, we have

identified new mechanisms in live mice that are not present or differ from those using the two above unphysiological approaches.

We have now compared the *in vivo* and *ex vivo* experiments using the same mouse models, imaging setup and imaging and analysis pipeline. For both experimental conditions, the focal plane was set at the level of the IHC nuclear region. We now mention this in the revised Methods section (ln. 474-476). Since the experimental parameters were kept similar among the different cochlear preparations, the reasons for the identified differences in Ca^{2+} dynamics and mechanisms are primarily related to the physiological status of the cochlea. This is further supported by the data showing that the characteristics of the Ca^{2+} waves recorded from cochlear explants closely matched those of the *in vivo* experiments performed following the opening of the cochlear partition (**Fig. 4**), indicating that the key difference is the maintenance of the physiological environment under *in vivo* conditions.

These new data are described across the revised Results section and have resulted in several new Figures.

I feel the manuscript would be improved by:

- Compiling these differences in a more organized manner

We believe that the largely revised manuscript is now making these points very clear.

- Making some attempt to explain why these differences would occur; and

Ideally, performing additional experiments could be performed to test some of these biological hypotheses. For instance, the EP could be measured in this preparation (to confirm that it is indeed being preserved, but also to see if changing the EP affects these Ca parameters). Or the ionic composition of the perilymph/endolymph could be altered. These are, of course, technically challenging, but would be necessary to connect the findings from this preparation to the large body of literature performed on *ex vivo* explants.

As mentioned above, we have performed several new experiments including the measurement of the EP and the *in vivo* recording following the opening of the membrane sealing the cochlear partition, which alters the ionic composition of the cochlear solutions.

- More thorough discussion about the impact of the imaging technique (imaging plane, analysis) on the measured parameters.

See same point above.

Question #2: What are the scientific conclusions/discoveries from this work?

The authors make some observations about changes in Ca transient/wave behavior over the course of development. In particular, they find that, in the second week, IHC transients are longer and smaller, and SC ICS waves are longer. However, they analyze these changes in different ways, sometimes showing day-by-day data (figure 1k and 3e) and sometimes aggregating by 1st vs 2nd postnatal week (figures 2e and 3d). The dichotomization between 1st and 2nd weeks seems somewhat arbitrary, especially since they only perform recordings from P4-P10, and the variance in certain parameters (fig 3e) between individual days appears to be as large, if not greater, than the variance between "weeks" (fig 3d).

As mentioned above, we have expanded the Results section and added a separate Discussion to highlight the above shortcomings. We completely agree that the separation between the 1st and 2nd week is somewhat arbitrary. Therefore, we have removed this separation in the revised version.

I feel the manuscript would be improved with:

- more consistent and/or rigorous statistical analysis of time-based trends

We have provided a clearer statistical analysis of the large number of comparisons, which have been included in the revised Results and/or Figure legends.

- clearer discussion of why these time-based trends may be occurring, informed by previous abundant literature on the evolution of these phenomena over this time period.

As mentioned above, the separation between the 1st and 2nd week has been removed. However, we have discussed the finding in a more coherent manner in the new separate Discussion.

The much more potentially significant finding from this work relates to the relationship between SC ICS waves and IHC transients. It is generally thought that IHC transients are coordinated directly as a result of SC ICS waves (through mechanisms involving extracellular ATP, K⁺ efflux, and depolarization of IHCs near these ATP release sites). However, it remains possible that the IHC Ca transients are coordinated (and have propagating waves) through mechanisms intrinsic to the IHCs themselves. In this study, the authors observe that the IHC waves propagate faster and farther than the SC waves. This implies that there is some IHC-intrinsic mechanism for their coordinated activity, and could represent a significant shift in how we understand how spontaneous, coordinated IHC activity occurs in late cochlear development. It is confusing, however, when they also conclude later that the "key role exerted by the Ca waves from SCs... in coordinated activation of the postsynaptic afferent terminals." The synaptic bouton activity was only correlated with respect to IHC transient activity, not adjacent SC activity.

As mentioned in **point 2** of the above "Summary of key aspects of our revision" (see Pgs. 1-2), the finding on the relation between SCs and IHCs is only one of the several new mechanisms identified from this first *in vivo* cochlear functional study.

We thank this Reviewer for spotting the confusing sentence about the control of SGN terminal activity by the synchronised IHCs, which we have now corrected and expanded (ln. 285-293). It is the synchronised activity of the IHCs, which is initiated by the Ca²⁺ waves from the supporting cells, that increases the recruitment of SGN terminals.

Overall, this potentially significant finding feels under investigated in this study. For example, can the authors shed any light on why this is happening? Is it an IHC-intrinsic propagation? Or is this happening because of extracellular spread of some paracrine signal (ATP, K), and/or different thresholds for triggering a Ca increase in a particular cell type? There are many questions.

This has been a very challenging project and in this first study we have identified several new mechanisms occurring *in vivo*. We completely understand the need to fully explain all of them, but unfortunately this will require several years of additional work because of the complexity of the system.

For the faster travelling of Ca²⁺ signals in the IHCs, we have proposed a possible mechanism in the new Discussion (ln. 352-354), although a detailed dissection of the underlying molecular mechanisms will require a lot of extra work and importing of specific transgenic mice targeting the function of both IHCs and SCs.

I feel the manuscript would be improved by:

- Additional analyses/experiments that describe more thoroughly what the relationship is between IHC and SC waves. Figures 3j-k show this in aggregate, it is unclear how frequently the IHC/SC waves co-occur overall. Also, in 3j/k: how many mice were tested? how many events/mouse? Why are there different numbers of events quantified in these two plots (3j: distance; 3k: speed)? Use of drugs or experimental manipulations to try to assess these connections more precisely would be difficult, but necessary to be able to make strong conclusions about the relationship between ICS waves and IHC transients.

We have performed additional analysis and highlighted some interesting observations for future experiments: **a)** the presence of "skipped" IHCs during their synchronised activity (ln. 152-158) and **b)** calculated the amount of Ca²⁺ waves able to trigger synchronised activity in the IHCs (ln. 214-217). Regarding the experimental manipulation, as highlighted in **point 1** of the above "Summary of key aspects of our revision" (see Pg. 1), *in vivo* manipulation requires transgenic mice targeting proteins thought to be involved in the observed mechanisms. Therefore, we performed new experiments using the two transgenic lines we hold at Sheffield (*Cav1.3* and *Otof* knockout mice). We have also listed the several additional experiments using the tools/mouse models at our disposal to provide a more in-depth understanding of the validity of the model used. For example, we have: **a)** provided a more direct comparison between the *in vivo* and *ex vivo* results, **b)** disrupted the physiological environment of the cochlear partition *in vivo* and found that supporting cells are able to generate large Ca²⁺ waves, similar to those recorded in cochlear explant, which do not occur under *in vivo* physiological conditions. Being able to identify the specific mechanisms linking SC and IHC

activity is in our future plans, but this will require importing different animal models and a few years of additional work.

We have also added the missing information in the Figure legend (now **Fig. 5**).

- At the very least, in the absence of additional experiments, a thorough discussion of this observation and its relationship to prior findings is required.

As highlighted in **points 1 & 2** of the above "Summary of key aspects of our revision" (see Pgs. 1-2), we have performed a large number of experiments and also provided a discussion with the above point explained.

Reviewer #3:

De Faveri and colleagues developed and described a novel method that utilize high resolution Ca²⁺ multiphoton imaging enabling the visualizing of calcium signals from different cellular subtypes in the cochlea, from anesthetized prehearing mice in vivo. This newly developed approach keeps the cochlear integrity (although still partially invasive) to investigate the molecular mechanisms underlying cochlear development and refinement for the auditory circuitry, here focusing on the Ca²⁺ waves occurring spontaneously in the organ of Corti.

The authors use different mouse models to express the genetically encoded Ca²⁺ indicator GCaMP6 in different cell types including inner hair cells (IHCs), supporting cells of the greater epithelial ridge (GER) and the bouton endings of the type-I auditory nerve fibers (ANFs). They demonstrate that IHCs, GER cells and type-I ANF bouton terminals display spontaneous Ca²⁺ signals in a more or less coordinated fashion, whereby Ca²⁺ waves are spontaneous generated by supporting cells in the GER, synchronizing the activity of several IHCs that at the end depolarize the type-I ANFs.

The approach used here is derived from a method already described, by the laboratory of Anthony Ricci (Stanford University). Secondly, the main result showing postnatal calcium waves in the organ of Corti in different cell types and how this coordinates, has also been shown extensively before, in excised cochlear tissue, in vitro (Bergles lab, Marcotti lab, Mammano lab and Gale lab). However, showing the known phenomenon of postnatal cochlear calcium waves in live animals, provides a 'quantum jump' step forward, providing high assurance that the phenomenon is actually occurring and of high significance in the living animal. This approach therefore provides a major breakthrough and will allow to gain future new insights in molecular mechanisms that sustain cochlear developments, nerve fiber refinement as well as mechanism that could participate to setting the encoding properties of the ANFs (modiolar vs pillar).

Thank you. Just one clarification: our in vivo approach is very different from that used by Ricci and colleagues on adult mice. More importantly, it was developed completely independently because our preliminary data were used for Marcotti's Wellcome Trust grant application well before the publication from the Ricci's lab. Nevertheless, it is important to have multiple approaches from different labs, all of which are likely to produce important results.

Overall, this study is well designed, straightforward with solid results and properly discussed in respect to the available literature. Comments provided here are minor:

Minor comments:

1) Supplementary figures with some raw traces of Ca²⁺ imaging recordings from IHCs, SCs and ANFs before off-line image treatment (i.e. removing out-of-focus frames, frames due to animal breathing, ...) should be added, for the reader to understand what to expect from such raw data.

This is an excellent suggestion. We thought that a visual representation of the videos before and after the off-line processing for the images would be the best approach to give the reader an appreciation of the image processing approach. Therefore, we have modified three videos to include both the pre- and post-processed images side-by-side: Ca²⁺ transients in the IHCs (**Supplementary Movie 1**), Ca²⁺ waves in the supporting cells (**Supplementary movie 3**) and Ca²⁺ signals in the SGNs (**Supplementary Movie 6**).

We have also included an example of a fluorescence trace from an IHC before and after the off-line image correction pipeline in the revised **Fig. 1j**. We hope that this proposed solution is satisfactory.

2) I am not questioning the validity of the Ca²⁺ imaging from ANFs but I am wondering if authors have observed different patterning of Ca²⁺ signals throughout the 2 weeks of development as during this period there is a lot of fiber pruning and refinement. Can authors discuss if they believe their approach could identify branching fibers vs individual fibers? Does the proportion of modiolar vs pillar fibers change over these 2 weeks?

Unfortunately, our approach cannot distinguish between branched and individual afferent fibres, since this is primarily a morphological feature. Even if we detect two or more simultaneous events, we cannot determine whether these are from the same of branching fibres.

SGN terminal recordings *in vivo* are extremely challenging and for this initial study we have pooled together all recordings between P4 and P9 (pillar and modiolar). Currently, our data suggests that this segregation in the SGN terminals seems to be already present during the first two postnatal weeks, indicating that it is likely to be genetically encoded, rather than activity dependent. We have added this speculation in the revised Discussion (ln. 369-376). Investigating specific changes in the distribution and activation characteristics of SGNs present on the pillar and modiolar side of the IHCs is a project on its own and is going to take at least another year of work.

3) p values, stars showing significance, and statistical tests are needed on graphs/figure legends.

Thank you for highlighting this point because we realised some of the values were missing in the submitted manuscript. To avoid any duplications, all the above information is now provided in the text and/or in the Figures/Figure legends.

4) Please provide more explicit details (in the method section) regarding animal age range for each mouse line.

We have now expanded the Methods section to include the age ranges used for the different mouse lines (ln. 409-425).

5) Line 93: Suggest to also include Babola et al., 2020 and 2021 showing P2Ry1 expression in the supporting cells of the GER.

Done

6) Line 96: Suggest to add the following reference regarding Ca²⁺ waves and cochlear tissue damage: Nowak et al., 2021: <https://doi.org/10.1523/ENEURO.0383-21.2021>

Done

7) Supplementary movie 6 is not listed in the main text.

Thank you for spotting this oversight. All movies are now listed.

REVIEWERS' COMMENTS

Reviewer #1 (Remarks to the Author):

The authors did a very good job addressing my concerns.

The new experiments with two knockout mouse lines, the new analyses, and the clarifications have strengthened the manuscript immensely.

I have no further comments.

Reviewer #2 (Remarks to the Author):

Thank you very much for this significant revision of your important manuscript. I appreciate the limitations of the prior format and feel that expansion of the manuscript enabled you to present the study in a much more thorough and organized manner. I feel that you have adequately addressed both of my major concerns (providing clear evidence of the differences between the in vivo and ex vivo techniques; and demonstrating the scientific discoveries made). This in vivo technique can be considered the gold standard against which ex vivo measurements (made previously as well as in the future) should be compared. Though there are obviously many more questions that are raised than answered in your study (in particular, the clear demonstration that IHC Ca transients propagate faster and farther than SC waves), this should serve as a foundation for much future work.

I only have one fairly minor request. Much of the data presentation and statistical comparison is made between a large number (hundreds to thousands) of IHCs/SC waves/boutons pooled across 10-30 mice (i.e. comparisons made in Fig 2d,f,g, 4l,m, 6e,j, and 7e; and timecourse presentations made in 2a,3d,4e-g). The effects, when analyzed at the individual cell/wave/bouton level, are large and therefore believable; however, it is necessary to provide some level of animal-level analysis to confirm these effects. For example, in Figure 4m, the difference between In vivo and In vivo_broken seal is statistically significantly different by testing at the wave level (N=621 vs 329 waves, respectively), but the animal number (N=10 vs 6, respectively) should be accounted for. This can be done by performing additional statistical testing, or by separating out the individual-animal data.

Reviewer #3 (Remarks to the Author):

The authors have greatly extended their manuscript with more experiments including control experiments from acutely excised cochlear tissues as well as from their in vivo preparation after seal disruption. These additional experiments improve the quality of this study and further highlight the importance of the in vivo approach.

In summary, the authors have addressed reviewer's concerns in a satisfactory manner.

Here are some suggestions for additional minor edits:

Line 418: please correct typo in the following sentence [...] allowed the investigation of Ca²⁺ singnals in supporting [...]

Methods section: It is not clear which jGCaMP8 isoforms were used in which cell type in this study. Did authors use jGCaMP8m for IHC imaging and jGCaMP8f for SGN imaging in KO animals? If yes, why do the Method section lines 451-452: "pGP-AAV-syn-jGCaMP8m-WPRE (#162375-AAV9) and AAV-syn-jGCaMP8f (#162376-AAV9)" mention both GCaMP8 under the Synapsin promoter? In that case, GCaMP8m and GCaMP8f will be expressed in nerve fibers. Have GCaMP8m and GCaMP8f been both tested in the set of SGN Ca²⁺ imaging? Please clarify.

Can the authors please provide details about the AAV used to express jGCaMP8m in hair cells beside the AAV capsid, like the promoter and source?

Reviewer #4 (Remarks to the Author):

In vivo spontaneous Ca²⁺ activity in the pre-hearing mammalian cochlea

Reviewer #1:

The authors did a very good job addressing my concerns.

The new experiments with two knockout mouse lines, the new analyses, and the clarifications have strengthened the manuscript immensely.

I have no further comments.

Thank you

Reviewer #2:

Thank you very much for this significant revision of your important manuscript. I appreciate the limitations of the prior format and feel that expansion of the manuscript enabled you to present the study in a much more thorough and organized manner. I feel that you have adequately addressed both of my major concerns (providing clear evidence of the differences between the *in vivo* and *ex vivo* techniques; and demonstrating the scientific discoveries made). This *in vivo* technique can be considered the gold standard against which *ex vivo* measurements (made previously as well as in the future) should be compared. Though there are obviously many more questions that are raised than answered in your study (in particular, the clear demonstration that IHC Ca transients propagate faster and farther than SC waves), this should serve as a foundation for much future work.

Thank you

I only have one fairly minor request. Much of the data presentation and statistical comparison is made between a large number (hundreds to thousands) of IHCs/SC waves/boutons pooled across 10-30 mice (i.e. comparisons made in Fig 2d,f,g, 4l,m, 6e,j, and 7e; and timecourse presentations made in 2a,3d,4e-g). The effects, when analyzed at the individual cell/wave/bouton level, are large and therefore believable; however, it is necessary to provide some level of animal-level analysis to confirm these effects. For example, in Figure 4m, the difference between *In vivo* and *In vivo_broken seal* is statistically significantly different by testing at the wave level (N=621 vs 329 waves, respectively), but the animal number (N=10 vs 6, respectively) should be accounted for. This can be done by performing additional statistical testing, or by separating out the individual-animal data.

We fully appreciate this point. We have now added a new Supplementary Figure showing the same mean data presented in the manuscript (Fig. 2a,2d,2f; Fig. 3d; Fig. 4e,4f,4g,4l,4m; Fig. 6d,6e,6j) but averaged by mouse (Supplementary Figure 1).

As previously mentioned in the Results section, for Fig. 2f and 2g (*ex vivo* data), the number of mice usable was small since in 1/3rd of them the IHCs had no spontaneous Ca²⁺ activity. Therefore, we decided to perform additional recordings from cochlear explants (*ex vivo*) of another 6 mice. These additional recordings further emphasised the very different dynamics of these *ex vivo* Ca²⁺ transients compared to those recorded *in vivo*. We found that in over 40% of the recordings, Ca²⁺ transients in the IHCs became progressively more frequent until the Ca²⁺ signal reached a sustained high-level lasting throughout the rest of the recording (see revised Fig. 2e). This behaviour, which was never observed *in vivo*, made it very difficult to have a reliable measure of the duration of the Ca²⁺ transients during these sustained high Ca²⁺ signals, preventing a meaningful comparison with the *in vivo* data. Therefore, we decided to better highlight this finding in the text (ln. 127-133 and revised Fig. 2e), but remove the previous panel g of Fig. 2 (half width of Ca²⁺ transients). We hope that this change is satisfactory.

Reviewer #3:

The authors have greatly extended their manuscript with more experiments including control experiments from acutely excised cochlear tissues as well as from their *in vivo* preparation after seal disruption. These additional experiments improve the quality of this study and further highlight the importance of the *in vivo* approach.

In summary, the authors have addressed reviewer's concerns in a satisfactory manner.

Thank you

Here are some suggestions for additional minor edits:

Line 418: please correct typo in the following sentence [...] allowed the investigation of Ca²⁺ signals in supporting [...]

Done

Methods section: It is not clear which jGCaMP8 isoforms were used in which cell type in this study. Did authors use jGCaMP8m for IHC imaging and jGCaMP8f for SGN imaging in KO animals? If yes, why do the Method section lines 451-452: "pGP-AAV-syn-jGCaMP8m-WPRE (#162375-451 AAV9) and AAV-syn-jGCaMP8f (#162376-AAV9)" mention both GCaMP8 under the Synapsin promoter? In that case, GCaMP8m and GCaMP8f will be expressed in nerve fibers. Have GCaMP8m and GCaMP8f been both tested in the set of SGN Ca²⁺ imaging? Please clarify.

Thank you for spotting this mistake in the Method section. As listed in the Result section, we used AAV9 from Addgene (AAV-syn-jGCaMP8f, #162376) for SGN transduction. However, we designed our own AAV-PHP.eB-jGCaMP8m with CMV promoter for the hair cells, which was packaged by Vector Builder. We have now amended the text (ln. 451-454).

Can the authors please provide details about the AAV used to express jGCaMP8m in hair cells beside the AAV capsid, like the promoter and source?

As mentioned above, we have now corrected the text and added the AAV used for the hair cells, including the promoter and source (ln. 451-454).

Reviewer #4:

Thank you for your time.